# How JEPA Avoids Noisy Features: The Implicit Bias of Deep Linear Self Distillation Networks

**Etai Littwin**    **Omid Saremi**    **Madhu Advani**
Vimal Thilak    Preetum Nakkiran    Chen Huang    Joshua Susskind

Apple

## Abstract

Two competing paradigms exist for self-supervised learning of data representations. *Joint Embedding Predictive Architecture* (JEPA) is a class of architectures in which semantically similar inputs are encoded into representations that are predictive of each other. A recent successful approach that falls under the JEPA framework is self-distillation, where an online encoder is trained to predict the output of the target encoder, sometimes using a lightweight predictor network. This is contrasted with the *Masked AutoEncoder* (MAE) paradigm, where an encoder and decoder are trained to reconstruct missing parts of the input in the data space rather, than its latent representation. A common motivation for using the JEPA approach over MAE is that the JEPA objective prioritizes abstract features over fine-grained pixel information (which can be unpredictable and uninformative). In this work, we seek to understand the mechanism behind this empirical observation by analyzing the training dynamics of deep linear models. We uncover a surprising mechanism: in a simplified linear setting where both approaches learn similar representations, JEPAs are biased to learn high-influence features, i.e., features characterized by having high regression coefficients. Our results point to a distinct implicit bias of predicting in latent space that may shed light on its success in practice.

## 1 Introduction

Representation learning has arguably been one of the most promising prospects and longest-standing goals of deep learning. In simple terms, representation learning refers to the process of automatically discovering and extracting meaningful features or representations from raw data. The learned representations can then be used to solve various downstream tasks, such as classification, and regression, or as general-purpose representations in robotics and embodied agents. In recent years, the practice of representation learning has witnessed remarkable strides, particularly in the domain of Self-Supervised Learning (SSL) [1, 2, 3, 4, 5, 6, 7, 8, 9, 10, 11, 12]. The SSL paradigm has garnered significant attention due to its ability to exploit vast amounts of unlabeled data, freeing practitioners from the burden of expensive annotation efforts. While many SSL approaches have been proposed in the literature, two paradigms have emerged as particularly successful, driving the bulk of the recent breakthroughs in the field:

**Masked AutoEncoders (MAE)**    The original MAE [8] and its successive variants (e.g. [13, 14]) introduced a training objective that seeks to reconstruct missing pieces of data from its partially masked input. This approach, while simple, has proven efficient and scalable [15]. Formally, the MAE objective uses an encoder function $f_W(X) : \mathbb{R}^d \to \mathbb{R}^d$ and a decoder function $g_V(f) : \mathbb{R}^d \to \mathbb{R}^d$ to predict a target $y$ given input $x$:

$$\mathcal{L}_{\text{MAE}} = \frac{1}{2}\mathbb{E}_{x,y \sim p(x,y)}\|g_V(f_W(x)) - y\|^2, \tag{1}$$

38th Conference on Neural Information Processing Systems (NeurIPS 2024).

where we define a joint distribution over inputs and targets $p = p(x, y)$ with $x, y \in \mathbb{R}^d$. Here we assume tied dimensions for simplicity.

In general, we can think of $x$ and $y$ as different inputs sharing the same underlying semantic information. One common practice is to let the input $x$ correspond to a partially masked input, and the target $y$ corresponds to the masked-out portions of the input. Note that a critical design choice of the MAE objective is that the loss penalizes the reconstruction error directly in the target's input space. This implies that, as perhaps some parts of the target $y$ cannot be predicted given $x$, the quality of the encoding $f_W$ crucially depends on how $x, y$ are sampled.

**Joint Embedding Predictive Architectures (JEPA)**    The JEPA family of models does not attempt to predict $y$ directly and instead opts to predict the latent representation of $y$ given by some encoding function. When the encoder function for $x$ and $y$ is shared, a simple JEPA objective is given by:

$$\mathcal{L}_{\text{JEPA}} = \frac{1}{2} \mathbb{E}_{x,y \sim p(x,y)} \| g_V(f_W(x)) - f_W(y) \|^2. \tag{2}$$

Unlike the objective in (1), the objective given by (2) is susceptible to collapse: it can be minimized trivially by employing an encoder and decoder that map all inputs to the same vector. Contrastive methods [16, 6, 17] avoid this issue by including an additional loss term that pushes apart representations for negative pairs (two different samples). The drawback to this approach is that it can require comparing a sample to many negative examples to work effectively. This has led to a rise in the popularity of non-contrastive SSL. With non-contrastive methods, it becomes imperative to implicitly or explicitly bias the representation mapping $f_W(x)$ to avoid its collapse to a trivial solution (e.g., $f_W(x) = 0, \forall x$). In this paper, we leverage the widely adopted [11] *StopGrad* operator, preventing the flow of gradients through the $f_W(y)$ branch. Prominent examples of this foundational architecture include BYOL [9], data2vec [10], SimSiam [11] and the more recent I-JEPA [18] and V-JEPA [19] models. Consequently, the JEPA models under consideration fall under non-contrastive self-distillation methods. We will refer to such methods as JEPA for the remainder of this paper.

While both paradigms have found practical use, characterizing the implicit bias of each remains a major research question. Recently, it was shown in [20] that in the vision domain, perceptual features mostly reside in the data subspace which explains a relatively small portion of the observed variance. This suggests that reconstruction losses are sub-optimal for perception tasks since they tend to prioritize learning subspaces that explain the bulk of the input variance. On the other hand, the JEPA approach appears to be more suitable for perception tasks: empirically, JEPA often achieves better downstream classification performance with fewer optimization steps [3, 19].

In this work, we seek to uncover the mechanisms behind these observations by analyzing tractable deep linear neural networks. To that end, we introduce a fully solvable toy model admitting a complete characterization of its training dynamics for both objectives. Our analysis is inspired by previous work on the greedy learning dynamics of gradient descent, and SSL in particular [21, 22, 23]. However, going beyond previous work, here we present a more nuanced analysis showing how the greedy nature of learning is affected by the choice of objective, model, and the properties of the data. Our analysis reveals a distinct implicit bias of JEPAs: the propensity of the JEPA objective towards learning influential features, defined as features with a large regression coefficient, a property for which the MAE objective is mostly agnostic. Moreover, we show that this distinct bias takes hold only when the encoder is deep and diminishes substantially when the encoder is shallow. This allows JEPAs to steer away from the top subspace of the data which explains its observed variance, and focus on the more semantically rich subspace. Our work therefore uncovers a fundamental implicit bias, shedding light on both the advantages and drawbacks of predicting in latent space.

To summarize, our specific contributions are:

1. We analytically solve the learning dynamics of JEPA and MAE, in the tractable theoretical setting of deep linear networks.

2. In this setting, we prove a significant difference between JEPA and MAE: informally, JEPA prioritizes "influential features" (features which are most informative in prediction), whereas MAE only prioritizes highly covarying features (even if they are noisy and thus less informative).

3. We show that our theoretical setting is rich enough to capture two more realistic generative processes, based on static and dynamic linear models.

Our results help clarify the precise advantages of joint-embedding architectures, and complement the recent results of [20] on limitations of reconstruction-based methods.

**Preliminaries**

In this paper, we consider linear over-parameterized encoders and a shallow linear decoder. This tractable setting allows the non-linear training dynamics [24, 25, 26, 27] of the deep linear encoder network to be solved exactly under a certain set of assumptions. Let $\{W^a\}_{a=1...L} \in \mathbb{R}^{d \times d}$ and $V \in \mathbb{R}^{d \times d}$ denote the set of weights for a linear encoder $f(x)$ of depth $L$, and that of a linear decoder $g(x)$ respectively, and consider the linear parameterization

$$\bar{W} = \prod_{a=1}^{L} W^a, \quad f(x) = \bar{W}x, \quad g(f(x)) = Vf(x), \tag{3}$$

where $\prod_{a=1}^{L} W^a$ indicates the conventional right-to-left product of matrices. As for the input data distribution, we will take $x, y$ to be samples from a centered distribution with covariances:

$$\mathbb{E}[xx^\top] = \Sigma^{xx}, \quad \mathbb{E}[yy^\top] = \Sigma^{yy}, \quad \mathbb{E}[xy^\top] = \Sigma^{xy}. \tag{4}$$

For the purpose of deriving learning dynamics, we will later assume that $\Sigma^{xx}, \Sigma^{xy}, \Sigma^{yy} \in \mathbb{R}^{d \times d}$ are diagonal. Let $SG(\bullet)$ denote the StopGrad operator applied on $\bullet$. Thus, the JEPA and MAE objectives become

$$\mathcal{L}_{\text{jepa}} = \frac{1}{2}\mathbb{E}_{x,y\sim p(x,y)}\|V\bar{W}x - SG(\bar{W})y\|^2, \quad \mathcal{L}_{\text{mae}} = \frac{1}{2}\mathbb{E}_{x,y\sim p(x,y)}\|V\bar{W}x - y\|^2. \tag{5}$$

We will denote the diagonal elements of the diagonalized covariance and correlations matrix by $\sigma_i^2 = \Sigma_{ii}^{xx}$ and $\lambda_i = \Sigma_{ii}^{yx}$ respectively, and let $\rho_i = \frac{\lambda_i}{\sigma_i^2}$ denote the regression coefficients.

To motivate our theoretical analysis, we next show empirically that depending on the particular values of $\{\lambda_i\}$ and $\{\rho_i\}$, optimizing the objectives in eq. (5) via gradient descent may learn different encoders entirely.

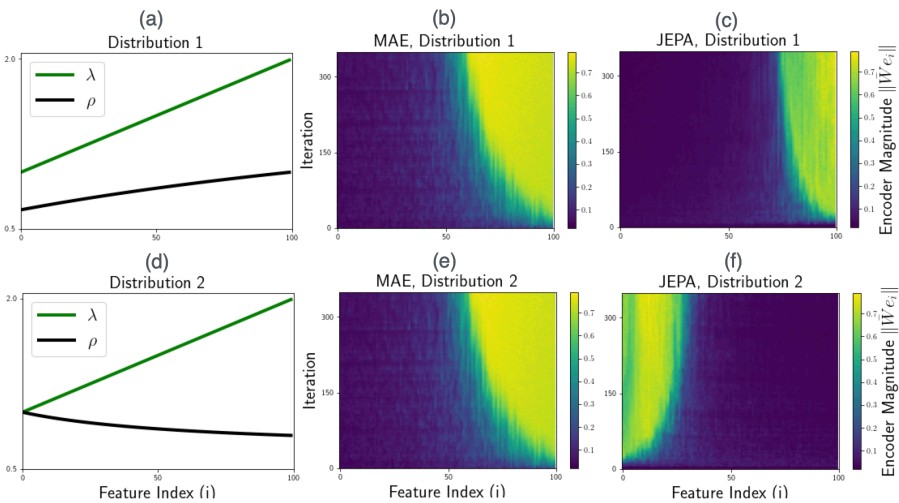

Figure 1: **Deep linear model trained using the MAE and JEPA objectives eq. (5).** Features indices ($x$-axis) are organized s.t. $\lambda_i$ is monotonically increasing and $\rho_i$ is (a) - (c) monotonically increasing or (d) - (f) monotonically decreasing. (b),(c): Both objectives learn features in the same order given distribution 1. In (e),(f) the MAE objective maintains the same learning order as in (b) on distribution 2, however, the JEPA objective reverses the learning order, due to sensitivity to $\rho_i$.

## 2 The Implicit Bias of JEPA

Before presenting the theoretical results, we describe a simple experiment using the setup in section 1 which will flesh out a surprising key difference between the JEPA and MAE objectives in eq. (5).

First, note that each coordinate $x_i$ of the input data represents an independent feature characterized by the covariance and regression coefficient $\lambda_i, \rho_i$, which we refer to as *feature parameters*. Let $e_i \in \mathbb{R}^d$ denote the $i$-th standard basis vector. We say that an encoder $\bar{W}$ has learned the $i$-th feature if its projection on $e_i$, namely $\|\bar{W}e_i\|$ is large. Intuitively, one should expect that if $\rho_i$ is small, $\|\bar{W}e_i\|$ should be small post training as well. Note that this should not be overly surprising since, at least in the case of the MAE objective, the global minimizer is achieved with the regression coefficients $V\bar{W} = \Sigma^{yx}(\Sigma^{xx})^{-1}$. In this investigation, however, we seek an understanding of the dynamics of training, by studying how the MAE and JEPA objectives prioritize different features in training, based on their feature parameters.

To this end, we track how the projections $\|\bar{W}e_i\|$ evolve during training for each component $i$ by training models on Gaussian data with feature parameters $\lambda_i, \rho_i$ according to the following two distributions (see Figure 1(a) and (d) for an illustration):

1. **Distribution 1**: Set both $\{\lambda_i\}, \{\rho_i\}$ to be monotonically increasing (i.e $\forall_{i>j}, \lambda_i > \lambda_j, \rho_i > \rho_j$).

2. **Distribution 2**: Set $\{\lambda_i\}$ to be monotonically increasing, and $\{\rho_i\}$ to be monotonically decreasing.

For each distribution, we initialize all the weights with a Gaussian initialization and train the models using both the MAE and JEPA objectives using gradient descent on batches of randomly sampled data. We illustrate the dynamics of training by measuring the magnitude of the encoder components $\|\bar{W}e_i\|$ during training in Figure 1. On the first distribution, we observe clear greedy learning dynamics for both objectives, where components with a higher $\lambda_i$ are learned earlier on in training. This behavior in greedy settings is in line with the findings in [21] which showed the same behavior in similar settings. However, on the second distribution, we observe a distinction: while the MAE objective retains its order of learning as in distribution 1, the JEPA objective exhibits a complete inversion of the learning order where features with a higher $\rho_i$, rather than a high $\lambda_i$ are learned first. This implies that in practical scenarios with a finite training budget, JEPA and MAE objectives can potentially learn entirely different features with little to no overlap, as illustrated in Figure 1. To better understand these phenomena, we turn to a dynamical analysis of a simple model that fully recovers our observations.

## 3   Dynamical Analysis

In the following theoretical analysis, we train the corresponding models employing objectives eq. (5) by optimizing all weights simultaneously starting at some initial weights configuration $\{W^a(0)\}_{a=1...L}$ and $V(0)$, using the gradient flow equations

$$\forall_a, \ \dot{W}^a = -\nabla_{W^a}\mathcal{L}, \quad \dot{V} = -\nabla_V\mathcal{L}, \tag{6}$$

where $\mathcal{L}$ is either JEPA or MAE objective. Generically, the two systems of ODEs are intractable without any simplifying assumptions. A special class of initializations discussed below permits analytical treatment of the problem, allowing us to derive a depth separation result indicative of differences in the inductive bias of the two objectives. In the following, we describe our set of assumptions on the encoder and decoder weights at initialization, and data distribution formally:

**Assumption 3.1.**

1. $\Sigma^{xx}, \Sigma^{yx}$ are diagonal and positive definite.

2. All weight matrices $W^a(0), a = 1...L, V(0)$ are initialized as $W^a = \epsilon^{\frac{1}{L}}U^a, V(0) = \epsilon^{\frac{1}{L}}U^v$ where $\{U^a\}_a, U^v$ are orthogonal matrices, and $0 < \epsilon \ll 1$ is an initialization scale.

3. For JEPA, we set $U^v$ to be diagonal, and for MAE we set $U^v$ such that $U^v(\prod_a U^a)$ is diagonal.

A few notes on assumption 3.1 before stating our results. The assumption on orthogonal weights implies that the weights are "balanced" at initialization, as is typically assumed in prior works on deep linear networks [28, 22]. The extra technical constraint on the uniformity of the initialization eigenvalues is necessary to deal with the non-uniformity of the data eigenvalues and approximately

holds in typical initializations with large Gaussian matrices. Likewise, we initialize $V$ differently for JEPA and MAE to achieve alignment of eigenvectors between the model and data at initialization. Although somewhat unrealistic, our special initialization schemes will prove informative to the general case. Applying assumption 3.1 to JEPA and MAE objectives in eq. (5) respectively, and using the gradient flow equations in eq. (6), one arrives at the following theorem:

**Theorem 3.2** (ODE Equivalence). *Suppose $\{W^a\}_{a=1\cdots L}$ and $V$ are initialized according to assumption 3.1. Let $\bar{w}_i = \|\bar{W}e_i\|$, where $e_i$s are the standard basis. Furthermore, assume the JEPA objective in eq. (5) is optimized using gradient flow according to eq. (6). Then, we have*

$$JEPA : \dot{\bar{w}}_i(t) = \bar{w}_i(t)^{3-\frac{1}{L}}\lambda_i - \bar{w}_i(t)^3\lambda_i\rho_i^{-1}. \tag{7}$$

*Similarly, the MAE objective eq. (5) is optimized using gradient flow according to eq. (6) yielding:*

$$MAE : \dot{\bar{w}}_i(t) = \bar{w}_i(t)^{2-\frac{1}{L}}\lambda_i - \bar{w}_i(t)^3\lambda_i\rho_i^{-1}. \tag{8}$$

Remarkably, the only difference between the JEPA and MAE dynamical equations is in an exponent. For a derivation of these dynamical equations see appendix B.1. Although the assumption 3.1 enables us to characterize the learning dynamics fully, we will show via numerical simulations (see section 4), that the observations made in this paper will not change qualitatively under the more general and realistic initialization. Since ODEs for each component $i$ decouple, for the remainder of this paper, we drop the subscript $i$ and consider a single ODE for $\bar{w}$ parameterized by the encoder and data feature parameters $\{\lambda, \rho\}$. A direct corollary to theorem 3.2 is that the training dynamics of MAE for $L >> 1$ as described in eq. (8) approach those of JEPA in eq. (7) for $L = 1$. Additionally, a depth-dependent difference is apparent in the fixed-point solutions between the two objectives. This is formalized in the following corollary:

**Corollary 3.3.** *Let $\bar{w}_{MAE}(t, L), \bar{w}_{JEPA}(t, L)$ denote the solutions to eqs. (7) and (8) for depth $L$ encoders and at time $t$, given initial condition $\bar{w}(0) = \epsilon$, we have*

$$\bar{w}_{MAE}(\infty, L) = \rho^{\frac{L}{L+1}}, \quad \bar{w}_{JEPA}(\infty, L) = \rho^L. \tag{9}$$

*In addition, it holds that the dynamics of a 1-layer JEPA model match an infinite-depth MAE*

$$\lim_{L'\to\infty} \bar{w}_{MAE}(t, L') = \bar{w}_{JEPA}(t, 1). \tag{10}$$

When $L > 1$, JEPA will suppress the encoding of noisier directions with a lower regression coefficient and this suppression increases with the network depth. At large depths, the encoder becomes low-rank to the point a single dominating eigenvalue corresponding to the feature with the largest regression coefficient remains. Note that corollary 3.3 also indicates that the JEPA solution for $L > 1$ is not reachable by the MAE objective, and vice versa. While interesting in itself, the difference between the two objectives runs deeper than their fixed-point solutions and is found by analyzing the training dynamics.

In the small initialization regime, the evolution of the weights during training for both objectives exhibits incremental learning dynamics, in which the encoder learns features progressively. We define the *critical time*, denoted as $t^*$, to be the time it takes for the encoder projection $\bar{w}$ to reach a positive finite-but-close-to-1 fraction $p$ of its final fixed point value. It is a quantity we wish to track since it captures an important data-dependent difference between JEPA and MAE training dynamics, concerning the order in which feature learning proceeds in the encoder:

$$\bar{w}(t^*) = p\bar{w}(\infty). \tag{11}$$

The dependence of this definition of the critical time on a choice for $p$ introduces a degree of arbitrariness in the actual value of $t^*$. However, as long as $p$ is not too close to 1 or 0, the leading order in the Laurent expansion for $t^* = t^*(\epsilon, \lambda, \rho)$ in $\epsilon$, as we show in section 3.1, is not affected by the specific values of $p$. The temporal ordering of $t^\star(\lambda, \rho)$s across all features, allows us to observe which feature takes priority in learning according to each objective.

## 3.1 Critical Time Analysis

To derive the critical time we first solve the dynamical equations in eqs. (7) and (8) in the following. The JEPA dynamics given by eq. (7) can be solved implicitly in a closed form as given by theorem B.7 with a full proof in the supplementary material section.

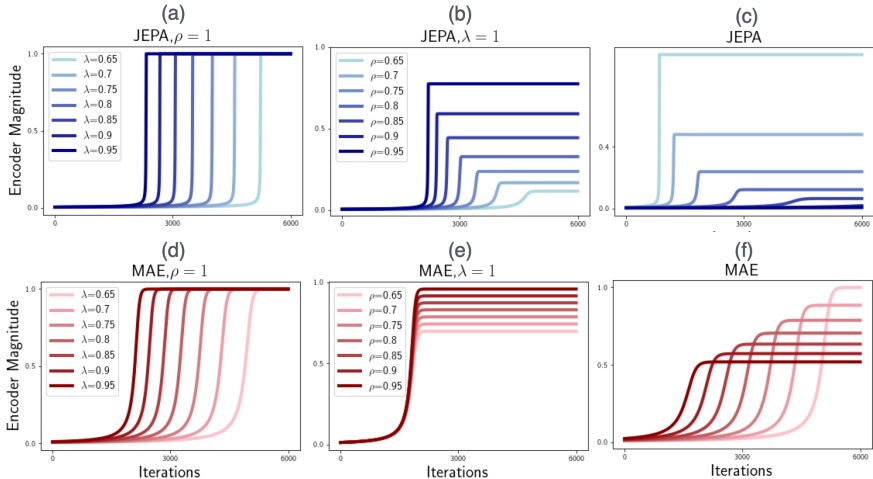

Figure 2: **Simulations of the JEPA and MAE equivalent ODEs (eqs. (7) and (8)).** Each curve represents a numerical simulation of the corresponding ODE, for different values of $\rho, \lambda$. (a), (d) darker curves correspond to higher $\lambda$ and $\rho = 1$. (b), (e) darker curves correspond to higher $\rho$ and $\lambda = 1$. As can be seen, both objectives exhibit greedy learning dynamics with respect to $\lambda$, however, only JEPA exhibits greedy dynamics with respect to $\rho$. (c), (f) darker curves correspond to higher $\lambda$ but lower $\rho$. In this case, the order of learning is inverted between the objectives due to the different trends in $\rho, \lambda$.

The closed form solution implicitly describes the full trajectory of $\bar{w}(t)$ for any time $t$. From this solution, it is clear that the encoder projection in the standard basis $\bar{w}$, starting from its initial value $\epsilon$ at $t = 0$, will reach its corresponding asymptotic value at $t = \infty$. The critical time $t^*$, defined in (11), is the time scale for which $\bar{w}$ exits the dynamical region near $t = 0$ and reaches a finite fraction of its asymptotic value. Theorem 3.4 gives the critical time for JEPA with arbitrary depth $L$ encoders:

**Theorem 3.4** (JEPA critical time). *The critical time $t^*$ in the small initialization regime $\bar{w}(t = 0) = \epsilon \ll 1$ for JEPA is given by*

$$t^*_{jepa} = \frac{1}{\lambda} \sum_{n=1}^{2L-1} \frac{L}{n \rho^{2L-n-1} \epsilon^{\frac{n}{L}}} + \Theta\left[\log(\epsilon)\right], \tag{12}$$

*as long as $p$ is not too close, as defined in eq. (81), to zero or one.*

In the small initialization regime, the JEPA solution given in theorem B.7 then describes a step-wise learning process of features where each feature is learned in the time scale given by eq. (12). We also derive an analogous theorem B.10 applied to the MAE objective. For technical reasons, we assume $L > 1$ in the following and provide the equivalent theorems for $L = 1$ in Appendix B.4. From the MAE dynamics summarized in , eq. (12) we arrive at the following result for MAE critical time:

**Theorem 3.5** (MAE critical time). *The MAE critical time $t^*$ in the small initialization regime of $\bar{w}(t = 0) = \epsilon \ll 1$ and $L > 1$ is given by*

$$t^*_{mae} = \frac{L}{\lambda(L-1)\epsilon^{\frac{L-1}{L}}} + \Theta(1). \tag{13}$$

*as long as $p$ is not too close, as defined in eq. (109), to zero or one.*

## 3.2 Comparing Learning Dynamics: JEPA vs. MAE

theorem 3.4 and theorem 3.5 reveal a crucial distinction between the two objectives. To understand how each objective prioritizes features, we note the functional form of the leading orders in $\epsilon$ in the critical time $t^\star$. For MAE, we observe that the leading order term depends principally on the inverse of $\lambda$. Crucially, we note that the next to leading order term (NLO) is small in comparison for any encoder depth $L$. This implies step-wise learning dynamics where the step ordering is predominantly set by the feature covariance $\lambda$. Conversely, features with identical $\lambda$ will be learned in the same

timescale. In the case of JEPA, we observe that the leading order term is again principally a function of $\lambda$, however, the NLO term, which depends inversely on the regression coefficient $\rho$, is increasingly large with the encoder depth $L$. That is, features with identical $\lambda$ but with different $\rho$ will be learned in different timescales, where the feature with the highest regression coefficient is prioritized. Moreover, the priority towards large $\rho$ increases with depth. This implies a meaningful and distinct implicit bias of the JEPA objective towards learning features with a high regression coefficient, increasingly so with depth. This is summarized in the following corollary:

**Corollary 3.6.** *Let $t^*_{jepa}(\epsilon, \lambda, \rho), t^*_{mae}(\epsilon, \lambda, \rho)$ denote the critical time for the JEPA and MAE objectives given feature parameters $\epsilon, \lambda, \rho$. WLOG assume $\rho' > \rho$, let $\Delta_\rho = \frac{1}{\rho} - \frac{1}{\rho'}$. Then, it holds that*

$$\frac{t^*_{jepa}(\epsilon, \lambda, \rho)}{t^*_{jepa}(\epsilon, \lambda, \rho')} = 1 + \frac{2L-1}{2L-2}\Delta_\rho \epsilon^{\frac{1}{L}} + \Theta(\epsilon^{\frac{2}{L}}), \quad \frac{t^*_{mae}(\epsilon, \lambda, \rho)}{t^*_{mae}(\epsilon, \lambda, \rho')} = 1 + \Theta\left[\epsilon^{\frac{L-1}{L}}\right] \qquad (14)$$

We note the strong dependency on depth $L$ and $\Delta_\rho$ in the JEPA case through the term $\Delta_\rho \epsilon^{\frac{1}{L}}$ in eq. (14), indicating greedy learning dynamics with respect to $\rho$, a property absent in the case of MAE. We next turn to empirical simulations to test the validity of our theory, as well as its generalizability in less stringent settings.

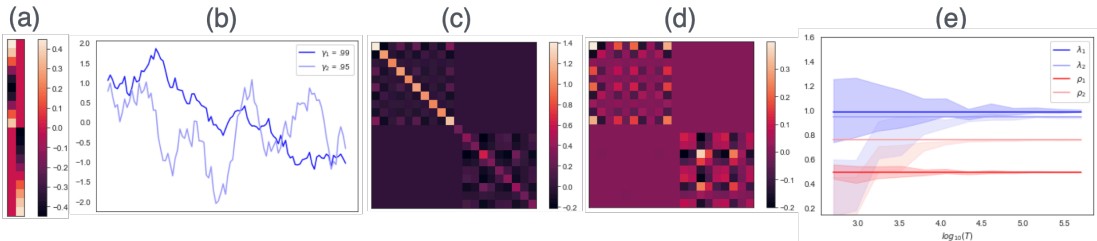

Figure 3: **Temporal model** (15): (a) $v_1, v_2$, (b) temporal dynamics of flickering $u$, autocorrelation $\gamma_1 = 0.99$ and $\gamma_2 = 0.95$, (c) $\hat{\Sigma}^{xx}$ empirical covariance with 500k samples, (d) $\hat{\Sigma}^{xy}$ empirical correlation, (e) predictions versus simulations of parameters $\lambda$ and $\rho$ varying $\log_{10}(T)$. $\lambda$ decreases from feature 1 to 2 whereas $\rho$ increase because noise added standard deviation decreases from 1 to 0.5. Note (e) show 2 standard deviation with 10 runs.

## 4 Numerical Experiments

We also conduct experiments directly verifying our theoretical findings. To that end, we numerically simulate the ODEs in eqs. (7) and (8) for different distributions and depths starting from a small initialization. In Figure 2, we fix the encoder depth $L = 5$ while varying the data parameters. As predicted by the theory, we observe a clear greedy learning process for both objectives. However, only the JEPA objective exhibits greedy dynamics with respect to $\rho$, where components with larger $\rho$ are learned first. Moreover, we observe that the JEPA objective can reverse the feature learning order relative to MAE when $\rho$ and $\lambda$ have the opposite trends, qualitatively reproducing the observation in section 2. In Figure 4, we run further simulations by varying the depth parameter $L$. As predicted, we observe that deeper encoders introduce a wider temporal spread between learning features with the same $\lambda$ but different $\rho$, in contrast to MAE which learns all features at the same time independent of the depth. Moving beyond gradient flow and the assumptions on initialization, in Figure 5 we train randomly initialized deep linear neural networks using stochastic gradient descent on randomly sampled batches of Gaussian data with varying values for $\rho, \lambda$, using both objectives. As is evident, our results carry over to this setup as well, where greedy learning of features can be seen with an objective dependent ordering according to $\rho, \lambda$.

### 4.1 Linear Generative Models

We consider generative models which satisfy our simultaneously-diagonalizable assumption (in the large sample limit). This is a setting in which our theory holds exactly and datasets with this property allow both JEPA and MAE models to learn the same modes, although not necessarily in the same

order. We analyze the simplest linear generative models which still lead to non-trivial learning differences between JEPAs and MAEs for two settings: *random masking* and a *temporal model*. The full details of these different settings can be found in Appendix C. In the first case, our views are random masks of the data, this corresponds to a static setting and we derive closed-form expressions for $\lambda$ and $\rho$ (see C.1). For the remainder of this section, we will focus in on the temporal model due to its simplicity and intuitive interpretation.

The temporal model considers the scenario where our views $x, y$ correspond to our data $z^1, ..., z^T$ at two consecutive times: $x = z^t$ and $y = z^{t+1}$. The goal is to predict the next frame from the previous one (this setting is consistent with comparing VideoMAE and V-JEPA video models). We now define a simple linear model corresponding to combining independently time-varying images $\{v^a\}$ to the model simultaneously with random noise. We can write this mathematically as:

$$z_i^t = \sum_a u^a(t) v_i^a + \xi_i^t, \quad t = 1, ... T. \tag{15}$$

Note that we are allowing the amplitude of the images to vary with time according to a scalar function $u(t)$; we make this choice to study the simplest non-trivial temporal variability. The full details of the setup and derivation for simultaneous diagonalizability can be found in Appendix C.2, where we derive the correlation coefficient for each feature:

$$\lambda_a = \gamma_a \|v^a\|^2, \quad \rho_a = \frac{\gamma_a \|v^a\|^2}{\sigma_a^2 + \|v^a\|^2}. \tag{16}$$

Here $\sigma_a$ represents the noise amplitude applied to mode $a$. See Figure 3 for a demonstration of our theory matched to simulations and an example of higher correlation modes with lower correlation coefficient in Figure 3(e). Additionally, a simulation demonstrating an approach to simultaneously diagonalizable empirical covariance and correlation $\hat{\Sigma}^{xx}$ and $\hat{\Sigma}^{xy}$ can be found in Figure 6. Combining (16) with our theory (14) demonstrates how JEPAs can be slower to learn noisy features in a way that MAEs are not. JEPA will also learn noisy features by converging to lower amplitude weights (9), making such features potentially less likely to be used downstream from the embedding.

## 5   Related Work

**Self-supervised learning.** One successful SSL paradigm is MAE [8] and its variants [13, 14] which learn representations via input space reconstruction. However, recent works (e.g., [20]) show that such generative paradigms often learn uninformative features for perception, since MAE-like methods tend to learn the data subspace that explains the bulk of observed variance and can include unhelpful information for perception. By contrast, the JEPA paradigm [9, 10, 11, 18, 19] learns to predict the representations of similar image views in the latent space. Such objectives have been found to prioritize semantic features over detailed pixel information, leading to superior performance for perception tasks. To prevent the feature collapse of JEPA, stop-gradients, and other architectural tricks are widely used. More recent works propose techniques like spectrum modulation [29] and weight regularization [30]. Another popular collapse prevention method is based on contrastive loss against negative image pairs [2, 5, 6, 11, 7]. In this work, we focus on MAE and JEPA methods, seeking to understand their implicit bias that leads to different training dynamics. To measure representation quality in JEPAs, [31] devised a metric that effectively counts the number of directions with a large regression coefficient in latent space. Our work can be seen as further motivation for this approach.

**Theoretical analysis of SSL.** There have been several recent works attempting to understand the success of non-contrastive SSL (JEPA) from various perspectives. [32] studies how the self-distillation in JEPA avoids representation collapse, finding the key role of eigenspace alignment between the predictor and input correlation matrix. The authors of [33] further provide a theoretical bridge between contrastive and non-contrastive objectives towards global and local spectral embedding methods respectively. It is shown that all these methods can be deployed successfully if the pairwise relations during SSL are aligned with the downstream task. While for MAE-based methods, [20] shows they tend to learn uninformative features by pixel reconstruction, unlike the JEPA objective that prioritizes semantic features. In this paper, we seek to understand the mechanism behind this empirical observation by characterizing the implicit bias of both methods using deep linear models.

**Learning dynamics in linear networks.** Deep linear neural networks have been used in [24, 25, 26, 27, 34, 28, 35, 36] to study the nonlinear dynamics of learning in various settings applying different assumptions, including the saddle point behavior as well as the step-wise behavior where distinct features are learned at different time scales. This is also in line with the observed greedy learning dynamics for modern architectures like Transformers [22, 23]. In the context of SSL, the authors of [21] observed a similar step-wise nature which corresponds to the learning of eigenmodes of the contrastive kernel $\Sigma^{yx} + \Sigma^{xy}$. However, we go beyond this observation by characterizing the distinctive implicit bias of JEPA relative to MAE, i.e., they learn the same features in a step-wise fashion but not necessarily in the same order. Surprisingly, this difference in behavior is only apparent when the encoder network is deep. Our findings about JEPA are related to empirical observations in [37] which show JEPA-based methods focus on "slow features" that vary slowly over time. This is confirmed in our studies where JEPA-based methods will prioritize "slow features" for which the marginal distribution has the smallest variance.

# 6   Limitations

Our theory has several clear limitations. The theoretical results presented in this paper are restricted to conditions in assumption 3.1, which include a deep linear MLP with some restrictions on the initialization scheme, as well as a simultaneously diagonalizable covariances $\Sigma^{yx}, \Sigma^{xx}$. It is worth discussing the implications of these assumptions, and the generalizability of our results to more typical settings. On the model side, empirical simulations with deep linear models initialized using the default initialization [38] indicate our results qualitatively generalize to popular initialization schemes. However, a potentially stronger assumption we make is simultaneously diagonalizability of $\Sigma^{yx}$ and $\Sigma^{xx}$, which allows JEPA and MAE to learn the same features. We provide generative frameworks that adhere to this property, however we expect data distributions of interest to significantly deviate from this assumption. Moreover, the generalizability of our claims to fully practical scenarios yet remains unexplored, and precise characterization of the implicit biases pertaining to more general data distributions and architectures is a direction for future work.

# 7   Discussion

We have introduced toy models of two of the most popular self-supervised algorithms, namely JEPA and MAE, where we can completely characterize the learning process. We have uncovered a novel learning principle that separates the two objectives: while MAE focuses on highly co-varying features early in training, the JEPA objective focuses on highly influential features (indicated by high values of the regression coefficient). This implicit bias of the JEPA objective allows it to focus on features that are predictive, yet contain minimal noise, as measured by their variance across the training set. This new understanding of the learning dynamics of JEPA sheds light on recent empirical observations and opens the door to new avenues of research strengthening these results. Certainly, one must first generalize these results to non-simultaneously-diagonalizable data distributions, as well as the more imposing challenge of non-linear models. Still, a less than rigorous leap to practical settings may still provide valuable insights. For example, an implicit bias towards predictive yet low-variance features may explain the tendency of JEPA to learn more semantic features, which are inherently less noisy. Conversely, it may shed light on JEPAs vulnerability to spurious or *slow* features. Additionally, the feature prioritization distinction between the objectives may bear on the efficiency of JEPA in learning semantic features quickly, as these features tend to reside in the low variance subspace of the data, as shown recently [20]. Finally, it is also worth investigating whether the mechanism uncovered here generalizes to other joint embedding architectures not relying on self-distillation.

# 8   Acknowledgments

We thank Dan Busbridge, Arwen Bradley, Stefano Cosentino, and Shuangfei Zhai for stimulating discussions and useful feedback on the research and writing of this paper.

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

# Contents

# Appendix A  Additional Figures

## A.1  Experimental Details

For Figure 1 and Figure 5, we train MLPs using stochastic gradient descent on random Gaussian data. We use a constant $d = 100$ width network and fix the encoder depth to be $L = 5$. We sample $x, y$ from a jointly Gaussian distribution with diagonal covariances, with different values of $\rho, \lambda$ per feature, as illustrated in the first column of each figure. At each iteration, we randomly sample a minibatch of 1000 samples and train for 5000 iterations.

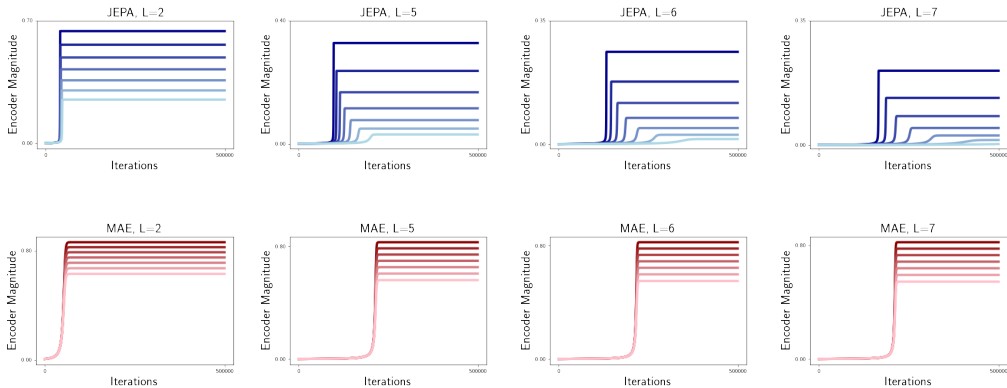

Figure 4: **Simulations of the JEPA and MAE equivalent ODEs for** $L = 2, 5, 6, 7$ **(eqs. (7) and (8)).** The covariance $\lambda$ is fixed to 1 across all curves, and darker curves correspond to a higher $\rho$. As evident, only in the case of the JEPA objective, deeper encoders induce a more pronounced incremental learning of features with respect to $\rho$.

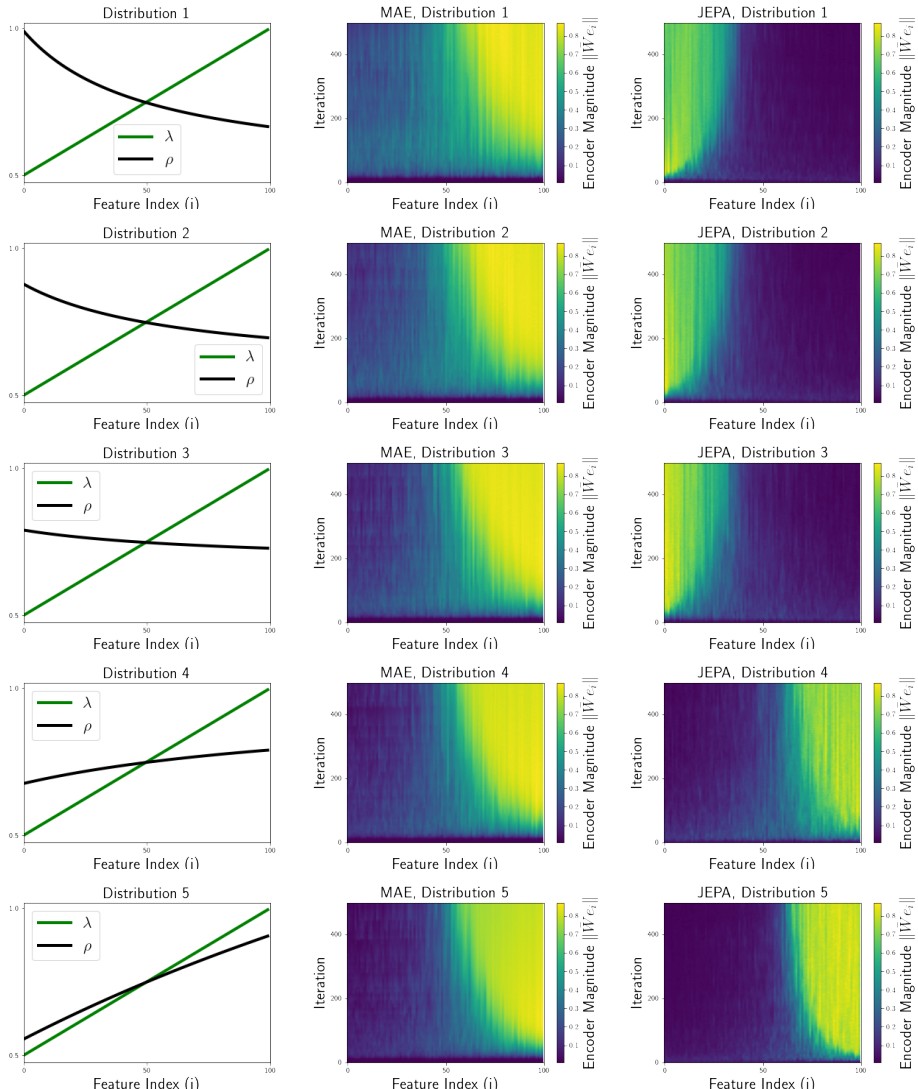

Figure 5: **Deep linear networks trained on Gaussian data.** The left most column represents the values of $\lambda, \rho$ used to generate the data. All networks were initialized using standard gaussian initialization with default scale, and the encoder depth is fixed to $L = 5$. The order of feature learning is dictated by $\rho$ for the JEPA objective, and $\lambda$ for the MAE objective.

# Appendix B  Proofs

In the following sections, we derive dynamical equations for JEPA and MAE and characterize their training dynamics. We also write down closed-form implicit solutions for their dynamics, all presented in the form of proofs for the theorems/corollaries mentioned in the main paper.

## B.1  Proofs of theorem 3.2 and corollary 3.3

**Theorem 3.2** (ODE Equivalence). *Suppose $\{W^a\}_{a=1\ldots L}$ and $V$ are initialized according to assumption 3.1. Let $\bar{w}_i = \|\bar{W}e_i\|$, where $e_i$s are the standard basis. Furthermore, assume the JEPA objective in eq. (5) is optimized using gradient flow according to eq. (6). Then, we have*

$$JEPA : \dot{\bar{w}}_i(t) = \bar{w}_i(t)^{3-\frac{1}{L}}\lambda_i - \bar{w}_i(t)^3\lambda_i\rho_i^{-1}. \tag{7}$$

*Similarly, the MAE objective eq. (5) is optimized using gradient flow according to eq. (6) yielding:*

$$MAE : \dot{\bar{w}}_i(t) = \bar{w}_i(t)^{2-\frac{1}{L}}\lambda_i - \bar{w}_i(t)^3\lambda_i\rho_i^{-1}. \tag{8}$$

*Proof.* Let us begin with the derivation of the JEPA dynamical equation. The gradient flow equations are given by

$$\forall_a, \quad \dot{W}^a(t) = -\nabla_{W^a(t)}\mathcal{L}_{jepa}, \quad \dot{V}(t) = -\nabla_{V(t)}\mathcal{L}_{jepa}. \tag{17}$$

Evaluating the gradient using the JEPA loss in eq. (5), one obtains

$$\forall_a, \quad \nabla_{W^a(t)}\mathcal{L}_{jepa} = \mathbb{E}\Big[\big(\prod_{b=a+1}^{L}W^b(t)\big)^\top V(t)^\top\big[V(t)\bar{W}(t)x - \bar{W}(t)y\big]x^\top\big[\prod_{b=1}^{a-1}W^b(t)\big]^\top\Big], \tag{18}$$

$$\nabla_{V(t)}\mathcal{L}_{jepa} = \mathbb{E}\Big[(V(t)\bar{W}(t)x - \bar{W}(t)y)x^\top\bar{W}(t)^\top\Big]. \tag{19}$$

Substituting for the expected values using $\mathbb{E}[xx^\top] = \Sigma^{xx}$ and $\mathbb{E}[yx^\top] = \Sigma^{yx}$, one gets

$$\forall_a, \quad \nabla_{W^a(t)}\mathcal{L}_{jepa} = \Big[\prod_{b=a+1}^{L}W^b(t)\Big]^\top V(t)^\top V(t)\bar{W}(t)\Sigma^{xx}\Big[\prod_{b=1}^{a-1}W^b(t)\Big]^\top \tag{20}$$

$$- \Big[\prod_{b=a+1}^{L}W^b(t)\Big]^\top V(t)^\top\bar{W}(t)\Sigma^{yx}\Big[\prod_{b=1}^{a-1}W^b(t)\Big]^\top,$$

$$\nabla_{V(t)}\mathcal{L}_{jepa} = \mathbb{E}\Big[(V(t)\bar{W}(t)x - \bar{W}(t)y)x^\top\bar{W}(t)^\top\Big] = V(t)\bar{W}(t)\Sigma^{xx}\bar{W}(t)^\top - \bar{W}(t)\Sigma^{yx}\bar{W}(t)^\top. \tag{21}$$

With the gradient flow equations eq. (17) in mind, we get the following system of ODEs for $\{w^a, V\}$ to solve

$$\forall_a, \quad \dot{W}^a(t) = \Big[\prod_{b=a+1}^{L}W^b(t)\Big]^\top V(t)^\top\bar{W}(t)\Sigma^{yx}\Big[\prod_{b=1}^{a-1}W^b(t)\Big]^\top \tag{22}$$

$$- \Big[\prod_{b=a+1}^{L}W^b(t)\Big]^\top V(t)^\top V(t)\bar{W}(t)\Sigma^{xx}\Big[\prod_{b=1}^{a-1}W^b(t)\Big]^\top,$$

$$\dot{V}(t) = \bar{W}(t)\Sigma^{yx}\bar{W}(t)^\top - V(t)\bar{W}(t)\Sigma^{xx}\bar{W}(t)^\top. \tag{23}$$

Similarly, using the MAE loss given by eq. (5), the gradient flow equations for MAE lead to the following system of ODEs

$$\forall_a, \quad \dot{W}^a(t) = \Big[\prod_{b=a+1}^{L}W^b(t)\Big]^\top V(t)^\top\Sigma^{yx}\Big[\prod_{b=1}^{a-1}W^b(t)\Big]^\top \tag{24}$$

$$- \Big[\prod_{b=a+1}^{L}W^b(t)\Big]^\top V(t)^\top V(t)\bar{W}(t)\Sigma^{xx}\Big[\prod_{b=1}^{a-1}W^b(t)\Big]^\top,$$

$$\dot{V}(t) = \Sigma^{yx}\bar{W}(t)^\top - V(t)\bar{W}(t)\Sigma^{xx}\bar{W}(t)^\top. \tag{25}$$

For ease of presentation, in this appendix we will use $V(t) = W^{L+1}(t)$, and denote $W(t) = W^{L+1}(t)\bar{W}(t)$. With this notation, the dynamical equations for JEPA can be rewritten for $W = W(t)$ and $\bar{W} = \bar{W}(t)$ as follows

$$\forall_a, \ \dot{W}^a(t) = \Big[ \prod_{b=a+1}^{L+1} W^b(t) \Big]^\top \bar{W}(t) \Sigma^{yx} \Big[ \prod_{b=1}^{a-1} W^b(t) \Big]^\top \tag{26}$$

$$- \Big[ \prod_{b=a+1}^{L+1} W^b(t) \Big]^\top W(t) \Sigma^{xx} \Big[ \prod_{b=1}^{a-1} W^b(t) \Big]^\top.$$

For $\dot{V}(t) = \dot{W}^{L+1}(t)$ we obtain

$$\dot{W}^{L+1}(t) = \bar{W}(t)\Sigma^{yx}\bar{W}(t)^\top - W(t)\Sigma^{xx}\bar{W}(t)^\top. \tag{27}$$

From the product rule for derivative, we have

$$\dot{W}(t) = \sum_{a=1}^{L+1} \Big[ \prod_{b=a+1}^{L+1} W^b(t) \Big] \dot{W}^a(t) \Big[ \prod_{b=1}^{a-1} W^b(t) \Big]. \tag{28}$$

Substituting the expression for $\dot{W}^a(t)$ in (28) gives

$$\dot{W}(t) = \sum_{a=1}^{L+1} \Big[ \prod_{b=a+1}^{L+1} W^b(t) \Big] \Big[ \prod_{b=a+1}^{L+1} W^b(t) \Big]^\top \bar{W}(t) \Sigma^{yx} \Big[ \prod_{b=1}^{a-1} W^b(t) \Big]^\top \Big[ \prod_{b=1}^{a-1} W^b(t) \Big] \tag{29}$$

$$- \sum_{a=1}^{L+1} \Big[ \prod_{b=a+1}^{L+1} W^b(t) \Big] \Big[ \prod_{b=a+1}^{L+1} W^b(t) \Big]^\top W(t) \Sigma^{xx} \Big[ \prod_{b=1}^{a-1} W^b(t) \Big]^\top \Big[ \prod_{b=1}^{a-1} W^b(t) \Big].$$

Similarly, the dynamical equations for MAE can be re-expressed for $W = W(t)$

$$\forall_a, \ \dot{W}^a(t) = \Big[ \prod_{b=a+1}^{L+1} W^b(t) \Big]^\top \Sigma^{yx} \Big[ \prod_{b=1}^{a-1} W^b(t) \Big]^\top \tag{30}$$

$$- \Big[ \prod_{b=a+1}^{L+1} W^b(t) \Big]^\top W(t) \Sigma^{xx} \Big[ \prod_{b=1}^{a-1} W^b(t) \Big]^\top.$$

For $\dot{V}(t) = \dot{W}^{L+1}(t)$ we get

$$\dot{W}^{L+1}(t) = \Sigma^{yx}\bar{W}(t)^\top - W(t)\Sigma^{xx}\bar{W}(t)^\top. \tag{31}$$

For MAE, plugging the expression for $\dot{W}^a(t)$ into (28) results in

$$\dot{W}(t) = \sum_{a=1}^{L+1} \Big[ \prod_{b=a+1}^{L+1} W^b(t) \Big] \Big[ \prod_{b=a+1}^{L+1} W^b(t) \Big]^\top \Sigma^{yx} \Big[ \prod_{b=1}^{a-1} W^b(t) \Big]^\top \Big[ \prod_{b=1}^{a-1} W^b(t) \Big] \tag{32}$$

$$- \sum_{a=1}^{L+1} \Big[ \prod_{b=a+1}^{L+1} W^b(t) \Big] \Big[ \prod_{b=a+1}^{L+1} W^b(t) \Big]^\top W(t) \Sigma^{xx} \Big[ \prod_{b=1}^{a-1} W^b(t) \Big]^\top \Big[ \prod_{b=1}^{a-1} W^b(t) \Big].$$

In the following, we study the relevant consequences of these dynamical equations and the assumption 3.1. Let us begin by noting that the above assumptions on initialization imply that the weights are "balanced" at initialization, i.e., $W^{a+1\top}(0)W^{a+1}(0) = W^a(0)W^{a\top}(0)$, for $a = 1...L$. In the following lemma, we show during training, the weights will remain balanced:

**Lemma B.1.** $W^{a+1\top}(t)W^{a+1}(t) = W^a(t)W^{a\top}(t)$ for $a = 1...L$, if $W^a(t)$ satisfies the ODE sets (26) and (27) or (30) and (31).

*Proof.* Multiplying both sides of the MAE dynamic equation (30) from the right by $W^{a\top}(t)$ for $a = 1 \cdots L - 1$ gives (after absorbing $W^{a\top}$ in the relevant products)

$$\dot{W}^a(t)W^{a\top}(t) = \Big[ \prod_{b=a+1}^{L+1} W^b(t) \Big]^\top \Sigma^{yx} \Big[ \prod_{b=1}^{a} W^b(t) \Big]^\top \tag{33}$$
$$- \Big[ \prod_{b=a+1}^{L+1} W^b(t) \Big]^\top W(t) \Sigma^{xx} \Big[ \prod_{b=1}^{a} W^b(t) \Big]^\top.$$

Rewriting (30) for $a + 1 \leq L$ layer index and multiplying by $W^{a+1\top}$ from the left results in

$$W^{a+1\top}(t)\dot{W}^{a+1}(t) = \Big[ \prod_{b=a+1}^{L+1} W^b(t) \Big]^\top \Sigma^{yx} \Big[ \prod_{b=1}^{a} W^b(t) \Big]^\top \tag{34}$$
$$- \Big[ \prod_{b=a+1}^{L+1} W^b(t) \Big]^\top W(t) \Sigma^{xx} \Big[ \prod_{b=1}^{a} W^b(t) \Big]^\top$$
$$= \dot{W}^a(t)W^{a\top}(t).$$

Transposing (34), one obtains

$$\dot{W}^{a+1\top}(t)W^{a+1}(t) = W^a(t)\dot{W}^{a\top}(t). \tag{35}$$

Using (27) and (31), it can also be shown that

$$V^\top(t)\dot{V}(t) = \dot{W}^L(t)W^{L\top}(t), \tag{36}$$
$$\dot{V}^\top(t)V(t) = W^L(t)\dot{W}^{L\top}(t).$$

Therefore, for $a = 1 \cdots L$

$$W^{a+1\top}(t)\dot{W}^{a+1}(t) + \dot{W}^{a+1\top}(t)W^{a+1}(t) = \dot{W}^a(t)W^{a\top}(t) + W^a(t)\dot{W}^{a\top}(t), \tag{37}$$

which can be written as

$$\frac{d}{dt}[W^{a+1\top}(t)W^{a+1}(t) - W^a(t)W^{a\top}(t)] = 0, \tag{38}$$

at all times. Consequently, $W^{a+1\top}(t)W^{a+1}(t) - W^a(t)W^{a\top}(t) = \mathcal{C}$ where $\mathcal{C}$ is a matrix with constant entries. On the other hand, according to assumption 3.1, $W^a$s $a = 1 \cdots L + 1$ are scaled orthogonal matrices at initialization, therefore $\mathcal{C} = \mathbf{0}$. This implies

$$W^{a+1\top}(t)W^{a+1}(t) = W^a(t)W^{a\top}(t), \tag{39}$$

at all times for $a = 1 \cdots L$, which proves the claim for MAE. The proof for the JEPA case is similar. This concludes the proof of the lemma. $\qquad\square$

Using the above lemma one can write the following alternative form for (29) and (32)

**Lemma B.2.** *The dynamical equations for JEPA and MAE share the same general form as follows*

$$\dot{W} = \sum_{a=1}^{L+1} [W(t)W(t)^\top]^{1-\frac{a}{L+1}} \Xi_{jepa/mae} [W(t)^\top W(t)]^{\frac{a-1}{L+1}} \tag{40}$$
$$- \sum_{a=1}^{L+1} [W(t)W(t)^\top]^{1-\frac{a}{L+1}} W(t)\Sigma^{xx}[W(t)^\top W(t)]^{\frac{a-1}{L+1}},$$

*where $\Xi_{jepa} = \bar{W}(t)\Sigma^{yx}$ and $\Xi_{mae} = \Sigma^{yx}$.*

*Proof.* Let $W^a(t) = Q_a(t)\omega_a(t)P_a(t)$ denote its SVD decomposition, where $Q^a$, $P^a$ are the left and right eigenbasis of the corresponding SVD decomposition. Using lemma B.1, we know

$$W^{a+1\top}(t)W^{a+1}(t) = W^a(t)W^{a\top}(t), \tag{41}$$

for $a = 1 \cdots L$. Substituting the corresponding SVD decomposition for $W^a$s implies

$$P_{a+1}^\top(t)\omega_{a+1}^2(t)P_{a+1}(t) = Q_a(t)\omega_a^2(t)Q_a^\top(t), \tag{42}$$

which leads to

$$P_{a+1}^\top(t) = Q_a(t), \forall_{a=1...L}, \tag{43}$$

and the conclusion that all $\omega$s are equal

$$\omega_a(t) = \omega(t), \forall_{a=1..L+1}. \tag{44}$$

It is easy to see that this implies

$$\Big[ \prod_{b=a+1}^{L+1} W^b(t) \Big] \Big[ \prod_{b=a+1}^{L+1} W^b(t) \Big]^\top = Q_{L+1}(t)\omega^{2(L-a+1)}Q_{L+1}^\top(t), \tag{45}$$

$$\Big[ \prod_{b=1}^{a-1} W^b(t) \Big]^\top \Big[ \prod_{b=1}^{a-1} W^b(t) \Big] = P_1(t)\omega^{2(a-1)}P_1^\top(t). \tag{46}$$

On the other hand

$$W(t)W^\top(t) = Q_{L+1}(t)\omega^{2(L+1)}(t)Q_{L+1}^\top(t), \tag{47}$$

$$W^\top(t)W(t) = P_1^\top(t)\omega^{2(L+1)}(t)P_1(t). \tag{48}$$

Using the above relations, the equations (29) and (32) result in the statement of the lemma. This concludes the proof. $\square$

Now we are in a position to state and prove the following theorem:

**Theorem B.3.** *For JEPA, suppose $\bar\omega(t) = \omega(t)^L$ with a diagonal $\omega(t)$. For any time t, the following ansatz*

$$\bar W(t) = \mathcal{U}\bar\omega(t), \tag{49}$$

$$W(t) = \mathcal{U}\bar\omega(t)^{1+\frac{1}{L}}, \tag{50}$$

*where $\mathcal{U}$ is a constant orthogonal matrix, solves dynamical equations for JEPA, if $\bar\omega(t)$ obeys the following differential equation*

$$\dot{\bar\omega}(t) = L\Big[\bar\omega(t)^{3-\frac{1}{L}}\Sigma^{yx} - \bar\omega^3(t)\Sigma^{xx}\Big], \tag{51}$$

*with $\bar\omega(0) = \epsilon\mathbb{1}$ at initialization.*

*Proof.* First note that the ansatz satisfies conditions 2 and 3 of the assumption 3.1 at initialization. Plugging the ansatz into (40) give

$$\mathcal{U}(1 + \frac{1}{L})\bar\omega^{\frac{1}{L}}(t)\dot{\bar\omega}(t) = (L+1)\mathcal{U}\Big[\bar\omega^3(t)\Sigma^{yx} - \bar\omega^{3+\frac{1}{L}}(t)\Sigma^{xx}\Big], \tag{52}$$

which is the claimed ODE after rearrangement of the terms and noting $\mathcal{U}$ is non-singular. Also, note that the equation for $V(t)$ is satisfied. In fact, from the ansatz one obtains

$$V(t) = W^{L+1}(t) = \mathcal{U}\bar\omega^{\frac{1}{L}}(t)\mathcal{U}^\top. \tag{53}$$

Substituting this in (27), we get

$$\mathcal{U}\frac{d\bar\omega^{\frac{1}{L}}(t)}{dt}\mathcal{U}^\top = \mathcal{U}\bar\omega^2(t)\Sigma^{yx}\mathcal{U}^\top - \mathcal{U}\bar\omega^{2+\frac{1}{L}}(t)\mathcal{U}^\top. \tag{54}$$

Rearranging the terms and expanding the left-hand side gives

$$\mathcal{U}\dot{\bar\omega}(t)\mathcal{U}^\top = L\mathcal{U}\Big[\bar\omega^{3-\frac{1}{L}}(t)\Sigma^{yx} - \bar\omega^3(t)\Sigma^{xx}\Big]\mathcal{U}^\top, \tag{55}$$

which is satisfied if the claimed ODE in the theorem is satisfied. This concludes the proof. $\square$

**Theorem B.4.** *For MAE, suppose $\bar{\omega}(t) = \omega^L(t)$ with a diagonal $\omega(t)$. The following ansatz*

$$\bar{W}(t) = \mathcal{U}\omega^L(t), \tag{56}$$

$$W^{L+1}(t) = \omega(t)\mathcal{U}^\top,$$

*where $\mathcal{U}$ is a constant orthogonal matrix, solves the dynamical equations* (40) *and* (31)*, if*

$$\dot{\bar{\omega}}(t) = L[\bar{\omega}^{2-\frac{1}{L}}(t)\Sigma^{yx} - \bar{\omega}^3(t)\Sigma^{xx}], \tag{57}$$

*with $\bar{\omega}(0) = \epsilon\mathbb{1}$ at initialization.*

*Proof.* Let us start by noticing that the ansatz is consistent with conditions 2 and 3 of the assumption 3.1 at initialization. From the ansatz

$$W(t) = W^{L+1}\bar{W}(t) = \omega^{1+L}(t). \tag{58}$$

Plugging this expression into the equation (40) gives

$$(L+1)\omega^L(t)\dot{\omega}(t) = \sum_{a=1}^{L+1} \omega^{2(L+1-a)+2(a-1)}\Sigma^{yx} - \sum_{a=1}^{L+1} \omega^{2(L+1-a)+L+1+2(a-1)}\Sigma^{xx}, \tag{59}$$

where the fact that $\Sigma^{yx}$ and $\Sigma^{xx}$ are diagonal was used. Simplifying the above expression, one obtains

$$\dot{\omega}(t) = \omega^L(t)\Sigma^{yx} - \omega^{2L+1}(t)\Sigma^{xx}. \tag{60}$$

One can rewrite this ODE in terms of $\bar{\omega} = \bar{\omega}(t)$. Multiply both sides of (60) $L\omega^{L-1}$(t) gives

$$\dot{\bar{\omega}}(t) = L[\bar{\omega}^{2-\frac{1}{L}}(t)\Sigma^{yx} - \bar{\omega}^3(t)\Sigma^{xx}]. \tag{61}$$

Now let us turn to (31). Using the ansatz eq. (56), we can write (31) as

$$\dot{\omega}(t) = \Sigma^{yx}\omega^L(t) - \omega^{L+1}(t)\Sigma^{xx}\omega^L(t) \tag{62}$$

which is the same as (60) given $\Sigma^{yx}$, $\Sigma^{xx}$ and $\omega$ are diagonal. This concludes the proof. $\square$

Keeping in mind $\bar{\omega}(0)$ is proportional to identity at initialization and $\Sigma^{xx}$ and $\Sigma^{xx}$ are assumed diagonal, $\bar{\omega} = \bar{\omega}(t)$ remains diagonal at all times during its evolution according to (51) and eq. (57). This also means the assumption $\omega = \omega(t)$ diagonal used during derivation of eq. (51) and eq. (57) is a self-consistent one. Let us denote $\bar{w}_i(t) = \bar{\omega}_{ii}(t)$. Observe that

$$\bar{w}_i(t) = \bar{\omega}_{ii}(t) = \|\bar{W}(t)e_i\|. \tag{63}$$

Using the above, one arrives at the following ODE for MAE

$$\dot{\bar{w}}_i(t) = \bar{w}_i(t)^{2-\frac{1}{L}}(\Sigma^{yx})_{ii} - \bar{w}_i(t)^3(\Sigma^{xx})_{ii}, \tag{64}$$

where the constant $L$ was absorbed into the definition of the gradient flow time. At this point, we can drop index $i$ to avoid clutter

$$\dot{\bar{w}}(t) = \bar{w}(t)^{2-\frac{1}{L}}\lambda - \bar{w}(t)^3\lambda\rho^{-1}. \tag{65}$$

Similarly, for JEPA one arrives at

$$\dot{\bar{w}}(t) = \bar{w}(t)^{3-\frac{1}{L}}\lambda - \bar{w}(t)^3\lambda\rho^{-1}, \tag{66}$$

where the definition of the regression coefficient $\rho$ was used. This completes the proof. $\square$

We may now proceed to prove corollary 3.3.

**Corollary B.5.** *Let $\bar{w}_{MAE}(t, L), \bar{w}_{JEPA}(t, L)$ denote the solutions to eqs.* (7) *and* (8) *for depth $L$ encoders and at time $t$, given initial condition $\bar{w}(0) = \epsilon$, we have*

$$\bar{w}_{MAE}(\infty, L) = \rho^{\frac{L}{L+1}}, \quad \bar{w}_{JEPA}(\infty, L) = \rho^L. \tag{9}$$

*In addition, it holds that the dynamics of a 1-layer JEPA model match an infinite-depth MAE*

$$\lim_{L' \to \infty} \bar{w}_{MAE}(t, L') = \bar{w}_{JEPA}(t, 1). \tag{10}$$

*Proof.* The fixed-point solutions for a finite $L$ can be easily derived by setting $\dot{\bar{w}} = 0$ for both the JEPA and MAE equivalent ODEs and solving for $\bar{w}$. To show that the limit of the MAE fixed-point at $L = \infty$ equals the JEPA fixed-point at $L = 1$, we note that the right-hand side of eq. (8) is a smooth function of $\zeta = \frac{1}{L}$ for $L > 0$. Hence, we have that $\bar{w}_{\mathrm{MAE}}(t, L = \infty)$ is the solution to

$$\dot{\bar{w}}(t, \infty) = \lim_{\zeta \to 0} \bar{w}(t)^{2-\zeta} \lambda - \bar{w}(t)^3 \rho^{-1} \lambda = \bar{w}(t)^2 \lambda - \bar{w}(t)^3 \rho^{-1} \lambda, \tag{67}$$

with the initial condition $\bar{w}(0)$, which is identical to the JEPA dynamics in eq. (7). $\qquad\square$

We will now proceed to discuss the results on JEPA dynamics next:

## B.2  JEPA dynamics: proofs of theorem B.7 and theorem 3.4

In this section, we solve the JEPA training dynamics given by the ODE in (66) for general depth $L$. Let us start by stating and proving the following lemma:

**Lemma B.6.** *Define*

$$\mathcal{I}_L(\psi) = \int \frac{d\psi}{\psi^{2L} - \psi^{2L+1}}, \tag{68}$$

*then*

$$\mathcal{I}_L = -\sum_{n=1}^{2L-1} \frac{1}{n\psi^n} + \log(\psi) - \log(1 - \psi) + \mathcal{C}, \tag{69}$$

*where $\mathcal{C}$ is a constant of integration.*

*Proof.* We use induction to prove (69). For $L = 1$, the integral is elementary given by

$$\mathcal{I}_1 = -\frac{1}{\psi} + \log(\psi) - \log(1 - \psi) + \mathcal{C}, \tag{70}$$

hence, the statement of the lemma holds. Next, we show if the statement of the lemma holds for an arbitrary $L \neq 1$, it will also hold for $L + 1$. Note that

$$\frac{1}{\psi^{2L+2}(1 - \psi)} - \frac{1}{\psi^{2L}(1 - \psi)} = \frac{1}{\psi^{2L+2}} + \frac{1}{\psi^{2L+1}}. \tag{71}$$

Integrating both sides with respect to $\psi$ gives the following recursion relation

$$\mathcal{I}_{L+1} - \mathcal{I}_L = -\frac{1}{(2L+1)\psi^{2L+1}} - \frac{1}{2L\psi^{2L}} + \mathcal{C}. \tag{72}$$

Assuming the statement of the lemma holds for some $L \neq 1$, one will have

$$\mathcal{I}_{L+1} = \mathcal{I}_L - \frac{1}{2L\psi^{2L}} - \frac{1}{(2L+1)\psi^{2L+1}} + \mathcal{C}, \tag{73}$$

$$= -\sum_{n=1}^{2L-1} \frac{1}{n\psi^n} + \log(\psi) - \log(1 - \psi) - \frac{1}{2L\psi^{2L}} - \frac{1}{(2L+1)\psi^{2L+1}} + \mathcal{C}, \tag{74}$$

$$= -\sum_{n=1}^{2(L+1)-1} \frac{1}{n\psi^n} + \log(\psi) - \log(1 - \psi) + \mathcal{C}. \tag{75}$$

Hence, the claimed relation (69) holds for $L + 1$ as well. An alternative proof strategy would have been based on integrating both sides of the following identity

$$\frac{1}{\psi^{2L}(1 - \psi)} = \frac{1}{1 - \psi} + \frac{\sum_{n=1}^{2L-1} \psi^n}{\psi^{2L}}. \tag{76}$$

This concludes the proof. $\qquad\square$

We are now in a position to state and prove the main theorem:

**Theorem B.7** (JEPA dynamics). *Suppose $\psi = \frac{1}{\rho} \bar{w}^{\frac{1}{L}}$. The JEPA dynamics given by eq. (7) admits the following implicit solution in $\psi$*

$$\psi(t) = \frac{\exp\left[\frac{\lambda}{L}\rho^{2L-1}t - \sum_{n=1}^{2L-1}\frac{1}{n-2L}\psi(t)^{n-2L} + \mathcal{C}\right]}{1 + \exp\left[\frac{\lambda}{L}\rho^{2L-1}t - \sum_{n=1}^{2L-1}\frac{1}{n-2L}\psi(t)^{n-2L} + \mathcal{C}\right]}. \tag{77}$$

*where $\psi(t=0) = \frac{1}{\rho}\epsilon^L$, and $\mathcal{C}$ is a constant of integration.*

**Theorem 3.4** (JEPA critical time). *The critical time $t^*$ in the small initialization regime $\bar{w}(t=0) = \epsilon \ll 1$ for JEPA is given by*

$$t^*_{jepa} = \frac{1}{\lambda} \sum_{n=1}^{2L-1} \frac{L}{n\rho^{2L-n-1}\epsilon^{\frac{n}{L}}} + \Theta\left[\log(\epsilon)\right], \tag{12}$$

*as long as $p$ is not too close, as defined in eq. (81), to zero or one.*

*Proof.* A change for variable of the form $\bar{w} = (\rho\psi)^L$ in the JEPA dynamical equation given by (66) leads to

$$\mathcal{I}_L = \int \frac{d\psi}{\psi^{2L} - \psi^{2L+1}} = \frac{\sigma^2\rho^{2L}}{L}t + \mathcal{C}. \tag{78}$$

To fix $\mathcal{C}$, we need to impose the initial condition at $t = 0$. Using lemma B.6 and noting $\psi_{|t=0} = \frac{1}{\rho}\epsilon^{\frac{1}{L}}$

$$\mathcal{C} = -\sum_{n=1}^{2L-1} \frac{\rho^n}{n\epsilon^{\frac{n}{L}}} + \frac{1}{L}\log(\epsilon) - \log(\rho) - \log(1 - \frac{1}{\rho}\epsilon^{\frac{1}{L}}). \tag{79}$$

To compute the critical time, take $\psi^* = \psi^*(t^*)$ to be the $\psi$ value at which we measure the critical time following the definition and arguments in (11). Putting (78) and (79) together leads to

$$\frac{\sigma^2\rho^{2L}}{L}t^* = -\sum_{n=1}^{2L-1}\frac{1}{n(\psi^*)^n} + \log(\psi^*) - \log(1-\psi^*) + \sum_{n=1}^{2L-1}\frac{\rho^n}{n\epsilon^{\frac{n}{L}}} - \frac{1}{L}\log(\epsilon) + \log(\rho) + \log(1 - \frac{1}{\rho}\epsilon^{\frac{1}{L}}). \tag{80}$$

As long as $\psi^*$ is not too close to 0 or 1 in an $\epsilon$-dependent way, meaning $\psi^* \sim \Theta(\epsilon^\beta)$ where $\beta < \frac{1}{L}$, or $1 - \psi^* \sim \Theta\left[\exp(-\epsilon^{-\beta})\right]$ where $\beta < \frac{2L-1}{L}$, the first three terms on the right-hand side in the above equation are sub-leading compared to the rest of the terms. Given that $\psi^* = p^{\frac{1}{L}}$, these conditions for $\psi^*$ translate to constraints on the scaling of $p$, defined in eq. (11). That is to say

$$p^{\frac{1}{L}} \sim \Theta(\epsilon^\beta) \text{ with } \beta < \frac{1}{L}, \quad 1 - p^{\frac{1}{L}} \sim \Theta\left[\exp(-\epsilon^{-\beta})\right] \text{ with } \beta < \frac{2L-1}{L}. \tag{81}$$

This implies the following Laurent expansion for the critical time holds

$$t^* = \sum_{n=1}^{2L-1} \frac{L\rho^{n-2L}}{n\sigma^2\epsilon^{\frac{n}{L}}} - \frac{1}{\sigma^2\rho^{2L}}\log(\epsilon) + \Theta(1). \tag{82}$$

Using $\rho = \frac{\lambda}{\sigma^2}$, the above equation can be rewritten as

$$t^*_{jepa} = \frac{1}{\lambda}\sum_{n=1}^{2L-1}\frac{L}{n\rho^{2L-n-1}\epsilon^{\frac{n}{L}}} + \Theta\left[\log(\epsilon)\right]. \tag{83}$$

This concludes the proof. $\square$

Embedded in the steps of the theorem's proof is an implicit closed-form solution to the JEPA dynamical equation in (66), which for completeness we state as a theorem below:

**Theorem B.7** (JEPA dynamics). *Suppose $\psi = \frac{1}{\rho}\bar{w}^{\frac{1}{L}}$. The JEPA dynamics given by eq. (7) admits the following implicit solution in $\psi$*

$$\psi(t) = \frac{\exp\left[\frac{\lambda}{L}\rho^{2L-1}t - \sum_{n=1}^{2L-1}\frac{1}{n-2L}\psi(t)^{n-2L} + \mathcal{C}\right]}{1 + \exp\left[\frac{\lambda}{L}\rho^{2L-1}t - \sum_{n=1}^{2L-1}\frac{1}{n-2L}\psi(t)^{n-2L} + \mathcal{C}\right]}. \tag{77}$$

*where $\psi(t=0) = \frac{1}{\rho}\epsilon^L$, and $\mathcal{C}$ is a constant of integration.*

*Proof.* Substituting $\bar{w} = (\rho\psi)^L$ in (66), one obtains

$$\int \frac{d\psi}{\psi^{2L} - \psi^{2L+1}} = \frac{\sigma^2}{L}\rho^{2L}t + \mathcal{C}. \tag{84}$$

From lemma B.6, we have

$$-\log(1-\psi) + \log\psi + \sum_{n=1}^{2L-1}\frac{1}{n-2L}\psi^{n-2L} = \frac{\sigma^2\rho^{2L}}{L}t + \mathcal{C}, \tag{85}$$

where

$$\psi_{|t=0} = \frac{1}{\rho}\epsilon^L, \tag{86}$$

$$\mathcal{C} = -\log(1-\psi_{|t=0}) + \log(\psi_{|t=0}) + \sum_{n=1}^{2L-1}\frac{1}{n-2L}\psi_{|t=0}^{n-2L}. \tag{87}$$

which, using the definition of the regression coefficient, can be rewritten as

$$\psi(t) = \frac{\exp\left[\frac{\lambda}{L}\rho^{2L-1}t - \sum_{n=1}^{2L-1}\frac{1}{n-2L}\psi(t)^{n-2L} + \mathcal{C}\right]}{1 + \exp\left[\frac{\lambda}{L}\rho^{2L-1}t - \sum_{n=1}^{2L-1}\frac{1}{n-2L}\psi(t)^{n-2L} + \mathcal{C}\right]}, \tag{88}$$

This completes the proof. $\square$

Now we move on to studying the MAE dynamics. It turns out, it makes sense to discuss MAE dynamics for $L = 1$ and $L > 1$ cases, separately:

## B.3   MAE dynamics: proofs of theorem B.10 and theorem 3.5 for $L > 1$

Now we turn to analyzing the MAE dynamics for an arbitrary encoder depth $L$. Let us start with the following lemma:

**Lemma B.8.** *Suppose $\Omega = [\delta, u]$, where $\delta$ is a constant and $u$, $\delta$ belong to $(0,1)$, then the following holds*

$$\mathcal{I}(u) = \int_\Omega \frac{d\psi}{\psi^2 - \psi^\alpha} = \frac{1}{(\alpha-2)u}\sum_{n=0}^\infty \frac{u^{n(\alpha-2)}}{n - \frac{1}{\alpha-2}} + \mathcal{C}, \tag{89}$$

*where $\alpha > 2$ and $\mathcal{C}$ is an integration constant.*

*Proof.* The integrand admits the following Laurent series expansion convergent within the integration domain $\Omega$

$$\frac{1}{\psi^2 - \psi^\alpha} = \frac{1}{\psi^2} + \sum_{n=1}^\infty \psi^{n(\alpha-2)-2}, \tag{90}$$

therefore, the series expansion can be integrated term by term

$$\mathcal{I}(u) = -\frac{1}{u} + \sum_{n=1}^\infty \frac{u^{n(\alpha-2)-1}}{n(\alpha-2)-1} + C_\delta \tag{91}$$

$$= \sum_{n=0}^\infty \frac{u^{n(\alpha-2)-1}}{n(\alpha-2)-1} + C_\delta \tag{92}$$

$$= \frac{1}{(\alpha-2)u}\sum_{n=0}^\infty \frac{u^{n(\alpha-2)}}{n - \frac{1}{\alpha-2}} + C_\delta, \tag{93}$$

where $C_\delta$ is some delta-dependent constant. In passing, we also recognize that the last expression above can be written in terms of the *Lerch transcendent function* $\Phi = \Phi(z, s, a)$ for the special value of $s = 1$

$$\mathcal{I}(u) = \frac{1}{(\alpha - 2)u} \Phi\left(u^{\alpha - 2}, 1, -\frac{1}{\alpha - 2}\right), \tag{94}$$

where $\Phi = \Phi(z, s, a)$ can be represented as a series over the complex plane

$$\Phi = \Phi(z, s, a) = \sum_{n=0}^{\infty} \frac{z^n}{(n + a)^s}. \tag{95}$$

It is convergent for $|z| < 1$ or on the unit circle $|z| = 1$ when $\Re(s) > 1$. This concludes the proof of the lemma. $\qquad\square$

Next, we proceed to state the main theorem of this section:

**Theorem 3.5** (MAE critical time). *The MAE critical time $t^*$ in the small initialization regime of $\bar{w}(t = 0) = \epsilon \ll 1$ and $L > 1$ is given by*

$$t^*_{mae} = \frac{L}{\lambda(L - 1)\epsilon^{\frac{L-1}{L}}} + \Theta(1). \tag{13}$$

*as long as $p$ is not too close, as defined in eq.* (109)*, to zero or one.*

*Proof.* Let us make the following change of the dependent variable in (64)

$$\bar{w}(t) = \rho^{\frac{L}{L+1}} u(t)^{\frac{L}{L-1}}. \tag{96}$$

Then it can be integrated to give the following

$$\mathcal{I} = \int \frac{du}{u^2 - u^{\frac{-3L+1}{1-L}}} = -\frac{\rho^{-\frac{1-L}{L+1}}\lambda(1 - L)}{L}t + \mathcal{C}, \tag{97}$$

where $\mathcal{C}$ is an integration constant. The integral $\mathcal{I}$ can be performed using lemma (B.8)

$$\mathcal{I} = \frac{L - 1}{(L + 1)u} \Phi\left(u^{\frac{L+1}{L-1}}, 1, \frac{1 - L}{1 + L}\right), \tag{98}$$

where $\Phi$ is the *Lerch transcendent function*. An initial condition needs to be imposed to fix the integration constant. Recall that

$$u_{|t=0} = \rho^{\frac{1-L}{L+1}} \epsilon^{\frac{L-1}{L}}. \tag{99}$$

Note that for $L > 1$

$$u_{|t=0} \to 0, \tag{100}$$

under the small initialization assumption where $\epsilon \to 0$. From (97)

$$\mathcal{C} = \mathcal{I}(u_{|t=0}). \tag{101}$$

Together with (100), we deduce

$$\mathcal{C} = \text{Leading Order}_{u \to 0} \left[\mathcal{I}(u)\right]_{|u = u_{|t=0}} + \Theta(\epsilon^r), \tag{102}$$

for some exponent $r$. Given

$$\frac{1}{u}\Phi(u^{\frac{L+1}{L-1}}, 1, \frac{1 - L}{1 + L}) = \frac{1 + L}{1 - L}\frac{1}{u} + \sum_{n=1}^{\infty} \frac{u^{\frac{L+1}{L-1}n - 1}}{n + \frac{1-L}{1+L}}, \tag{103}$$

This allows us to write

$$\frac{\Phi}{u} = \frac{1 + L}{1 - L}\frac{1}{u} + \Theta(u^{\frac{2}{L-1}}). \tag{104}$$

This fixes $\mathcal{C}$ to be

$$\mathcal{C} = -\frac{1}{\rho^{\frac{1-L}{1+L}} \epsilon^{\frac{L-1}{L}}} + \Theta(\epsilon^{\frac{2}{L}}). \tag{105}$$

To compute the critical time, we start with (97) and (105)

$$\frac{\rho^{-\frac{1-L}{L+1}}\lambda(1-L)}{L}t^* = -\frac{1}{\rho^{\frac{1-L}{1+L}} \epsilon^{\frac{L-1}{L}}} - \frac{L-1}{(L+1)u^*}\Phi(u_*^{\frac{L+1}{L-1}}, 1, \frac{1-L}{1+L}). \tag{106}$$

To further proceed, we make the following observation, stated as a lemma:

**Lemma B.9.** *The second term on the right-hand side of* (106) *is always sub-leading compared to the first unless $u_*$ is chosen so that $u_* \sim \Theta(\epsilon^\beta)$ or $1 - u_* \sim \Theta\left[\exp(-\epsilon^{-\beta})\right]$, where $\beta > \frac{L-1}{L}$.*

*Proof.* Singular points of $\mathcal{I} = \mathcal{I}(u)$ are at $u = 0$ or $u = 1$. The leading term in the Taylor expansion near $u = 0$ is

$$\mathcal{I}(u) = -\frac{1}{u} + \Theta(u^{\frac{2}{L-1}}). \tag{107}$$

If $u_* \sim \Theta(\epsilon^\beta)$ with $\beta < \frac{L-1}{L}$, then $\mathcal{I}(u_*)$ would be sub-leading compared to the first term on the right-hand side of (106). In addition, $\mathcal{I}(u)$ contains a logarithmically singular point at $u = 1$. Taylor-expanding around $u = 1^-$ gives

$$\mathcal{I} = -\frac{L-1}{L+1}\log(1-u) + \Theta(1). \tag{108}$$

If $1 - u_* \sim \Theta\left[\exp(-\epsilon^{-\beta})\right]$ then $\mathcal{I}(u_*) \sim \epsilon^{-\beta}$ which is sub-leading compared to the first term of the right-hand side of (106) for $\beta < \frac{L-1}{L}$. This completes the proof. $\qquad\square$

Therefore, for any choice of $u^*$, unless it is too close to either of the two singular points at $u = 0$ and $u = 1$ (as quantified in the above lemma), the $\mathcal{I}$ term in (106) is sub-leading. Given $u^* = p^{\frac{L-1}{L}}$ where $p$ is the fraction in the definition of the critical time, defined in eq. (11), the conditions on $u^*$ become constraints on $p$ not scaling too close to zero or one with small $\epsilon$

$$p^{\frac{L-1}{L}} \sim \Theta(\epsilon^\beta) \text{ or } 1 - p^{\frac{L-1}{L}} \sim \Theta\left[\exp(-\epsilon^{-\beta})\right], \text{ with } \beta < \frac{L-1}{L}. \tag{109}$$

This means in the limit $\epsilon \to 0$ the leading term in the Laurent expansion in $\epsilon$ of the critical time is given by

$$t^*_{mae} = \frac{L}{\lambda(L-1)\epsilon^{\frac{L-1}{L}}}. \tag{110}$$

The sub-leading $\Theta(1)$ contribution to the critical that depends on the choice of $u^*$

$$t^*_{mae} = \frac{L}{\lambda(L-1)\epsilon^{\frac{L-1}{L}}} + \frac{\gamma}{(L-1)\lambda} + \Theta(\epsilon^{\frac{2}{L}}), \tag{111}$$

where $\gamma = L\rho^{\frac{1-L}{1+L}}\mathcal{I}(u^*)$, which is $u^*$-dependent. This concludes the proof. $\qquad\square$

Let us state one of the intermediate results (closed-form solution to MAE dynamics) in proving the above theorem:

**Theorem B.10** (MAE dynamics). *Suppose $L > 1$ and let $\psi = \rho^{\frac{1-L}{L+1}}\bar{w}^{\frac{L-1}{L}}$. The MAE dynamics given by eq.* (8) *admits the following implicit solution in $\psi$*

$$t = -\frac{\rho^{\frac{1-L}{L+1}}\mathcal{C}L}{\lambda(L-1)} + \frac{\rho^{\frac{1-L}{L+1}}L}{\lambda(L+1)\psi(t)}\Phi\left(\psi(t)^{\frac{L+1}{L-1}}, 1, \frac{1-L}{1+L}\right) \tag{112}$$

*where $\Phi := \Phi(z, a, q)$ is the Lerch transcendent function defined by*

$$\Phi(z, s, a) = \sum_{n=0}^{\infty} \frac{z^n}{(a+n)^a}, \quad |z| < 1, \tag{113}$$

*and $\mathcal{C}$ is an integration constant.*

From eq. (112) we can extract the critical time:

*Proof.* To prove this theorem, note that (97) gives

$$-\frac{\rho^{-\frac{1-L}{L+1}}\lambda(1-L)}{L}t = -\mathcal{C} + \frac{L-1}{u(L+1)}\Phi\left(u^{\frac{L+1}{L-1}}, 1, \frac{1-L}{1+L}\right),\qquad(114)$$

which reduces to the statement of the theorem after rearranging terms and constants, where $\mathcal{C}$ is given by (105). This proves the theorem. $\qquad\square$

## B.4 MAE: shallow encoder $L = 1$

For a single-layer encoder we have (64)

$$\frac{1}{\sigma^2}\int_{\bar{w}}\frac{d\bar{w}(t)}{\rho\bar{w}(t) - \bar{w}^3(t)} = t + \mathcal{C},\qquad(115)$$

which has a closed-form solution

$$\ln[\bar{w}^2(t)] - \ln\left(|\bar{w}^2(t) - \rho|\right) = 2\sigma^2\rho t + \mathcal{C}.\qquad(116)$$

Solving for $\bar{w}(t)$ and imposing the initial condition leads to

$$\bar{w}^2(t) = (\rho - \bar{w}^2(t))\exp\left[2\lambda t + \mathcal{C}\right],\qquad(117)$$

$$\bar{w}(t) = \frac{\sqrt{\rho}\exp\left[\lambda t + \mathcal{C}/2\right]}{\sqrt{1 + \exp\left[2\lambda t + \mathcal{C}\right]}},\qquad(118)$$

$$\mathcal{C} = \log\left(\frac{\epsilon^2}{\rho - \epsilon^2}\right).\qquad(119)$$

Qualitatively, the dynamics of $\bar{w}(t)$ for $\epsilon \ll 1$ can be described as follows:

1. $\bar{w}(t)$ remains indistinguishable from zero until a critical time $t^* = t^\star(\epsilon)$ is reached.
2. At $t > t^\star(\epsilon)$, $\bar{w}(t)$ grows rapidly until it converges to its asymptotic value given by $\bar{w}(\infty) = \sqrt{\rho}$.

We can derive the critical time $t^*$ defined in (11). The solution (116) can be written

$$2\lambda t^* + \log\left(\frac{\epsilon^2}{\rho - \epsilon^2}\right) = \log(\bar{w}_*^2) - \log\left(|\bar{w}_*^2 - \rho|\right) \sim \Theta(1).\qquad(120)$$

The leading contribution to the critical time in the limit $\epsilon \to 0$ is

$$t^\star(\epsilon, \lambda) \approx \frac{|\log(\epsilon)|}{\lambda}.\qquad(121)$$

In summary, for $L = 1$ and small initialization, learning of projections of the encoder in the standard basis occurs in a step-wise manner over the time scale $t^*$ which is *controlled by $\lambda$ only*.

## B.5 Proof of corollary 3.6

**Corollary B.11.** *Let $t^*_{jepa}(\epsilon, \lambda, \rho), t^*_{mae}(\epsilon, \lambda, \rho)$ denote the critical time for the JEPA and MAE objectives given feature parameters $\epsilon, \lambda, \rho$. WLOG assume $\rho' > \rho$, let $\Delta_\rho = \frac{1}{\rho} - \frac{1}{\rho'}$. Then, it holds that*

$$\frac{t^*_{jepa}(\epsilon, \lambda, \rho)}{t^*_{jepa}(\epsilon, \lambda, \rho')} = 1 + \frac{2L-1}{2L-2}\Delta_\rho\epsilon^{\frac{1}{L}} + \Theta(\epsilon^{\frac{2}{L}}), \quad \frac{t^*_{mae}(\epsilon, \lambda, \rho)}{t^*_{mae}(\epsilon, \lambda, \rho')} = 1 + \Theta\left[\epsilon^{\frac{L-1}{L}}\right]\qquad(14)$$

*Proof.* The statement of the corollary is easily seen to hold if one forms JEPA/MAE critical time ratios using the results in theorem 3.4 and theorem 3.5) for some fixed $\lambda$ but two values of the regression coefficient, denoted as $\rho$ and $\rho'$. Taylor-expanding the resulting expression around $\epsilon = 0$ up to sufficient order proves the corollary. $\qquad\square$

## Appendix C  Linear Generative Models

### C.1  Random masking with isotropic noise

The framework we will use is to assume a data sample $z \in \mathbb{R}^d$ is generated as:

$$z_i = \sum_k s_k B_{ki} + \eta_i. \tag{122}$$

where $\eta_i$ is noise and $s_k$ are model coefficients for important factors (rows of $B$). We will also find it useful to define $\mathbb{E}\left[s^2\right] = \frac{1}{d}\sum_{k=1}^d \mathbb{E}\left[s_k^2\right]$ and will assume that the variance of these coefficients and noise is isotropic: $\mathbb{E}\left[\eta_i^2\right] = \mathbb{E}\left[\eta^2\right]$.

We then sample $x$ and $y$ from masked versions of $z$ as:

$$x_i = z_i m_i \quad y_i = z_i(1 - m_i) \quad m_i = \begin{cases} 1 & \text{with probability } f \\ 0 & \text{with probability } 1 - f \end{cases} \tag{123}$$

Under assumptions that $B_{ki} \sim \mathcal{N}(0, \frac{1}{d})$ and in the high dimensional limit of infinite $d$, infinite sample size and some assumptions on the distribution of model coefficients (for instance that they are $\gamma$-sparse with finite $\gamma$), we can show:

**Theorem C.1.** $\Sigma^{xx}$ and $\Sigma^{xy}$ are simultaneously diagonalizable in the basis $B$ and have diagonal elements:

$$\sigma_i^2 = f^2 \mathbb{E}\left[s_i^2\right] + f \mathbb{E}\left[\eta^2\right] + (f - f^2)\mathbb{E}\left[s^2\right], \tag{124}$$

$$\lambda_i = f(1 - f)(\mathbb{E}\left[s_i^2\right] - \mathbb{E}\left[s^2\right]). \tag{125}$$

*From this, we can see that*

$$\rho_i = \frac{\lambda_i}{\sigma_i^2} = \frac{(1 - f)(\mathbb{E}\left[s_i^2\right] - \mathbb{E}\left[s^2\right])}{\mathbb{E}\left[s^2\right] + \mathbb{E}\left[\eta^2\right] + f(\mathbb{E}\left[s_i^2\right] - \mathbb{E}\left[s^2\right])} \tag{126}$$

We can see from this that the order in which components are learned does not depend on the masking ratio, but it does impact the final solution learned and the precise timing in which components are learned. This derivation can also be applied to show that all the parameters in a sparse basis will eventually be learned, but the weighting will depend on the sparsity and noise level of the data and which architecture and loss function we choose.

In order to derive these results, we compute the covariances and correlation terms of this model since our results in the main paper are computed in terms of these:

$$\mathbb{E}\left[x_i x_j\right] = \mathbb{E}\left[m_i m_j\right] \mathbb{E}\left[z_i z_j\right] = \left((f - f^2)\delta_{ij} + f^2\right)\mathbb{E}\left[z_i z_j\right]. \tag{127}$$

It follows that:

$$\mathbb{E}\left[xx^T\right] = f^2 \mathbb{E}\left[zz^T\right] + (f - f^2)\text{Diag}(\mathbb{E}\left[zz^T\right]). \tag{128}$$

Diag($\bullet$) here is an operator removing all non-diagonal elements in the matrix $\bullet$. Similarly we can derive that $\mathbb{E}\left[m_i(1 - m_j)\right] = f(1 - f)(1 - \delta_{ij})$ and $\mathbb{E}\left[(1 - m_i)(1 - m_j)\right] = f(1 - f)\delta_{ij} + (1 - f)^2$ to show that:

$$\mathbb{E}\left[xy^T\right] = f(1 - f)\mathbb{E}\left[zz^T\right] - f(1 - f)\text{Diag}(\mathbb{E}\left[zz^T\right]), \tag{129}$$

$$\mathbb{E}\left[yy^T\right] = (1 - f)^2 \mathbb{E}\left[zz^T\right] + (f - f^2)\text{Diag}(\mathbb{E}\left[zz^T\right]). \tag{130}$$

Note that all of these matrices have the same form: a weighted sum of the full data covariance and the diagonal of this covariance. It is helpful therefore to compute this covariance directly:

$$\mathbb{E}\left[zz^T\right]_{\epsilon,s,m} = B^T \Sigma^s B + \sigma_\epsilon^2 I = B^T\left(\Sigma^s + \sigma_\epsilon^2 I\right)B \tag{131}$$

where $\Sigma^s \in \mathbb{R}^{d \times d}$ is a diagonal with components $\Sigma_{jk}^s = \mathbb{E}\left[s_k^2\right]\delta_{jk}$. Unfortunately, the diagonal of the data covariance matrix above is not generally going to be exactly simultaneously diagonalizable with the data covariance unless it is proportional to an identity matrix. We can however show that this will occur in a certain limit. To see this consider the diagonal elements of the full data covariance:

$$\mathbb{E}\left[z_i^2\right] = b_i^T\left(\Sigma^s + \sigma_\epsilon^2\right)b_i = \sum_k B_{ki}^2(s_k^2 + \sigma_\epsilon^2) \tag{132}$$

In the limit where $\sum_k B_{ki}^2 s_k^2$ becomes a self-averaged quantity meaning that it does not fluctuate as we draw new instances of $B$ and $s$ from the dataset, we will have that these diagonal elements are also the same for all $i$ and it is therefore proportional to an identity matrix.

To keep our derivation simple we will assume that the coefficients are $\gamma$-sparse, meaning $[\gamma \cdot d]$ are drawn from a distribution $P_s$ and the rest are zero ($[\bullet]$ rounds to the nearest integer). It follows that:

$$\lim_{\gamma d \to \infty} \mathbb{E}\left[z_i^2\right] = \mathbb{E}\left[s^2\right] + \sigma_\epsilon^2 \tag{133}$$

Thus, we have simultaneous diagonalizability.

## C.2   Temporal model

$$z_i^t = \sum_a u^a(t) v_i^a + \xi_i^t \tag{134}$$

This corresponds to combining a set of images $\{\mathbf{v}^a\}$ but flickering them with a time-varying intensity $u(t)$. We will assume the following properties in the large $T$ limit and then construct appropriate functions:

$$\mathbb{E}\left[(u^a(t))^2\right] = 1, \quad \mathbb{E}\left[u^a(t)u^a(t+1)\right] = \gamma_a, \quad \mathbb{E}\left[u^a(t)\right] = 0. \tag{135}$$

$$\mathbb{E}\left[u^a(t)u^b(t)\right] = 0, \quad \mathbb{E}\left[u^a(t)u^b(t+1)\right] = 0. \tag{136}$$

This will hold for instance if we generate

$$u^a(1) = \eta_0^a, \quad u^a(t+1) = \gamma_a u^a(t) + \sqrt{1-\gamma_a^2}\eta_t^a, \quad \eta_t^a \sim \mathcal{N}(0,1). \tag{137}$$

We must ensure $T$ is large enough to allow for good mixing of values for simulations. Note that the mixing time will depend on $\gamma_a$.

We will also assume that the noise is uncorrelated in time so that: $\mathbb{E}\left[\xi_i^t \xi_i^{t+1}\right] = 0$. It follows that in the large $T$ limit:

$$C_{ij}^{xy} = \sum_a \gamma_a v_i^a v_j^a \tag{138}$$

For the purposes of this experiment, we assume the images take up different parts of the input space so that we can easily tune the level of noise that shows up in each to explore the trade-offs between temporal correlation and noise. In this case, all vectors $v^a$ are orthogonal by definition so we immediately know that the above expression for $C^{xy}$ is diagonal. We then assume the noise is $\xi_i \sim \mathcal{N}(0, \sigma_{a(i)}^2)$ where $a(i)$ denotes the image that pixel $i$ belongs to.

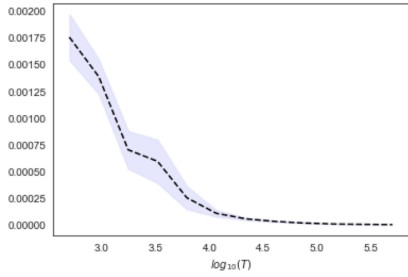

Figure 6: **Simultaneous diagonalizability demonstration**. We are in the same setting as Fig 3. Simultaneous diagonalizability is demonstrated by measured by mean of squared error in off-diagonal elements when attempting to diagonalize $\hat{\Sigma}^{xy}$ using the eigenbasis of $\hat{\Sigma}^{xx}$. The plot show 2 standard deviation with 10 runs.

We then have that

$$C_{ij}^{xx} = \sum_{a=1}^M v_i^a v_j^a + \sum_a \sigma_a^2 \delta_{ij} \delta_{a(i)a(j)} \tag{139}$$

Here $\delta_{ij}$ refers to the Kroneker delta function which is zero when $i \neq j$ and one otherwise. It follows that $v^a$ are all rescaled eigenvectors and these matrices are simultaneously diagonalizable with the corresponding parameters:

$$\lambda_a = \gamma_a \|v^a\|^2, \quad a = 1, ..., M, \quad \lambda_a = 0 \text{ otherwise.} \tag{140}$$

$$\rho_a = \frac{\gamma_a \|v^a\|^2}{\sigma_a^2 + \|v^a\|^2}, \quad a = 1, ..., M, \quad \rho_a = 0 \text{ otherwise.} \tag{141}$$

## Appendix D   Imagenet Experiments

Our results in the paper predict that JEPA will tend to focus on the lower subspace of data variance where most of the perceptual features reside as claimed in [20]. To test this prediction in a realistic setting, we use the empirical setup described in [20] to study the differences in the feature learning dynamics of I-JEPA [3] and MAE [8]. We first describe the empirical setup and then make our observations.

**Setup**   : We train a vision transformer (ViT) [39] encoder using I-JEPA [3] and MAE [8] on ImageNet-1K [40] data resized to $64 \times 64$ pixels. We use MAE-style masking to ensure parity with the masking function and copy all other hyperparameters described in I-JEPA [3] and MAE [8]. We build additional datasets by removing certain principal components of the full data subspace as described in [20] — **bottom** refers to the dataset that preservers bottom $25\%$ of explained variance while **top** denotes the dataset with top $75\%$ of explained variance. We use the original (**full**) images as well the filtered images described above and extract representations using encoders trained with I-JEPA and MAE. We then compare the representation between **full-bottom** and **full-top** via canonical correlation analysis (CCA) [41] for several checkpoints that are gathered during training.

**Observe**   that I-JEPA's features obtained from the full and bottom datasets show higher similarity compared to MAE throughout training while the trend is reversed for full and top images. This implies that JEPA learns features from the bottom portion of the PCA space **faster** than MAE. Balestrieo and Lecun [20] suggest that the bottom portion of the spectrum contain features useful for discrimination tasks which is what JEPA focuses on during optimization.

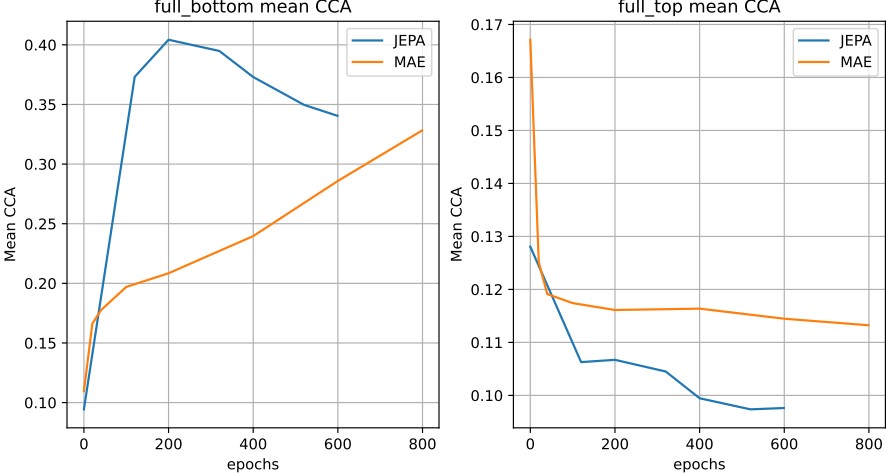

Figure 7: Feature similarity dynamics for I-JEPA vs MAE on Imagenet dataset.

## Appendix E   Broader Impact

This work is a theoretical contribution to shed light on techniques in self-supervised learning to try to understand their implicit biases. The analyses reveal differences in the kinds of features that get

learned, which can potentially be helpful to practitioners in selecting which technique to use when. These benefits may be important in deriving better models for a variety of downstream tasks that people care about such as classification, planning, and decision-making.

