# OpenReview forum: "How JEPA Avoids Noisy Features: The Implicit Bias of Deep Linear Self Distillation Networks"
_NeurIPS.cc/2024/Conference — NeurIPS 2024 poster_

### Official Review · Reviewer_MHjv · 2024-07-02

**Soundness:** 3
**Presentation:** 3
**Contribution:** 3
**Rating:** 8
**Confidence:** 2

**Summary:**

This paper presents a theoretical analysis of the JEPA and MAE SSL objectives for deep linear networks. Under a somewhat restrictive diagonal covariance assumption, the authors demonstrate that the critical time for learning a feature is dependent only on the input variance for MAE, while JEPA prioritizes learning features with high regression coefficients, which are predictive yet have minimal noise as measured by their variance across the training set (for large depth encoders).

The authors provide various numerical simulations to validate their theory and show that their findings are robust with respect to the network initialization scheme, even when using a more restrictive initialization necessary for the theory.

Overall, this paper offers valuable insights into the behavior of JEPA and MAE SSL objectives in deep linear networks, and provides a solid foundation for future research in this area.

**Strengths:**

- This work is the first to provide theoretical insights into the empirical observation that JEPA-based approaches tend to learn 'abstract' features more efficiently than MAE. Although the theory is limited to a restrictive case of diagonal linear networks, it calls for further research to generalize these findings.

- The numerical simulations and Section 3 effectively support the theory while also highlighting the qualitative differences between MAE and JEPA.

**Weaknesses:**

-While the paper is generally clear, the presentation of the theoretical results could be improved. Specifically, it would be helpful to expand on Theorems 4.4 and 4.6 in the main paper, as their validity is not immediately clear without referring to the appendix.

-In addition to the diagonal assumption, the authors focus on the case where the predictor/decoder is linear. However, it is important to use a deep predictor/decoder for both JEPA and MAE methods in practice.

-It would be useful to provide more intuition on what covariance_{x,x} and covariance_{x,y} represent for usual pretraining tasks. This would help readers better understand the underlying concepts and appreciate the significance of the theoretical results.

**Questions:**

- Would the main theoretical results hold for deep predictor/decoder?

- Given that MAE predicts a corrupted input in the same input space, what does it mean for covariance_{x,y} to differ from covariance_{x,x}, and how does the corruption affect this difference?

**Limitations:**

Authors adequately addressed the limitations.

---

> ### Author Rebuttal · Authors · 2024-08-07
>
> **Q.** Would the main theoretical results hold for deep predictor/decoder?
>
> **A.** Our analysis can be directly extended to deep linear decoders/predictors without changing the results in any qualitative way. Nonlinear predictors however are non-tractable, and therefore beyond the scope of this work. We opted to use a shallow rather than a deep linear predictor in our analyzed model due to 1) a deeper predictor would add to an already notation heavy presentation, without adding or changing the results in any meaningful way. 2) JEPA models typically use a lightweight predictor relative to the encoder, and even linear predictors (see [1]). And 3) Shallow linear predictors have been analyzed theoretically in [1] and [2] and have been shown to work even in practice (though not necessarily producing SOTA results).
>
> **Q.** Given that MAE predicts a corrupted input in the same input space, what does it mean for covariance_{x,y} to differ from covariance_{x,x}, and how does the corruption affect this difference?
>
> **A.** The difference comes from masking or mapping that is applied in the self-supervised model. In the case of random masking one place the difference occurs is on the diagonal elements: because the same element will always be zero in either x_i or y_i, the diagonal elements of C_{xy} will be zero, and this is not true of C_{xx}. For a full derivation and the full impact please see Appendix C.1. One intuitive impact of this difference is that increasing iid pixel noise will not impact diagonal elements of C_{xy} but will impact diagonal elements of C_{xx} leading to a suppression of regression coefficients.
>
> We thank the reviewer for their time and effort in reviewing our paper. If we have sufficiently addressed the reviewers concerns, we kindly ask them to consider raising their score.
>
> [1] Tian et al: Understanding Self-Supervised Learning Dynamics Without Contrastive Pairs“
> [2] Richemond et al: “The Edge of Orthogonality: A Simple View of What Makes BYOL Tick”

---

> > ### Comment · Reviewer_MHjv · 2024-08-08
> > **Thank your for the rebuttal.**
> >
> > The paper highlights a fundamental difference between two main SSL paradigm (MAE and JEPA).  By presenting empirical evidence and theoretical insights, it suggests that JEPA may be more effective for learning semantic features in images, but faces challenges with low-level features required for fine-grained tasks. The authors' rebuttal successfully addressed my initial concerns and provided additional supporting evidence from ImageNet, which strengthened the paper's contributions and led me to revise my score.
> >
> > I believe this work will be highly relevant and of interest to the SSL community. Thank you for your contribution!

---

### Official Review · Reviewer_5fn3 · 2024-07-11

**Soundness:** 3
**Presentation:** 3
**Contribution:** 3
**Rating:** 7
**Confidence:** 5

**Summary:**

The paper investigates the implicit bias of predictive self-supervised learning methods, specifically focusing on the Joint-Embedding Predictive Architecture (JEPA) and comparing it with the Masked Autoencoders (MAE). The study presents a theoretical analysis of the learning dynamics of these methods, revealing how different objectives lead to varied implicit biases in feature learning. It also includes numerical experiments with linear generative models to illustrate the theoretical findings.

**Strengths:**

- The paper provides a rigorous theoretical framework for understanding the implicit bias of JEPA and MAE. This is valuable for the community as it offers insights into why these methods might prioritize certain features over others during training.
- By comparing JEPA and MAE, the paper helps delineate the strengths and weaknesses of each method. This comparative approach can guide practitioners in choosing the appropriate method for their specific tasks.

**Weaknesses:**

-The paper would benefit from experiments on more diverse datasets, including real-world data. This would demonstrate the practical implications of the theoretical findings and validate their robustness in more varied scenarios.
- The numerical experiments are limited in scope. They primarily focus on linear generative models, which may not fully capture the complexities of real-world data. Expanding the experiments to include more diverse datasets and model architectures would strengthen the findings.

**Questions:**

- How well do the theoretical results generalize to non-linear models and more complex data distributions? The paper's findings are based on linear generative models, which may not fully represent the behavior of JEPA and MAE in practical settings.
- what are the practical implications of the implicit bias observed in JEPA and MAE? How should practitioners account for these biases when applying these methods to real-world tasks?
-The paper suggests that JEPA may be more efficient in learning semantic features. Can this efficiency be quantified in practical scenarios, and how does it impact downstream tasks such as classification and object detection?
-aree there strategies to mitigate the negative effects of the implicit bias observed in JEPA and MAE? For instance, can architectural modifications or additional regularization techniques help balance the feature learning dynamics?

**Limitations:**

The paper offers valuable theoretical insights into the implicit bias of JEPA and MAE in predictive self-supervised learning. However, it falls short in terms of related work, novelty,and experimental scope. Addressing these weaknesses and answering the important questions raised would significantly enhance the impact and applicability of the research.

---

> ### Author Rebuttal · Authors · 2024-08-07
>
> **Q.** “How well do the theoretical results generalize to non-linear models and more complex data distributions?”
>
> **A.** We have conducted additional experiments on ImageNet (in attached pdf), which are consistent with aspects of our theoretical predictions. (Please see the pdf for details of the setup).
> Regarding our focus on linear models: The question of how to quantify the efficiency of learning semantic features in practical scenarios may require different metrics than the ones used in this paper. Beyond analytical tractability, concentrating on deep linear networks allows us to separate the question of which features are learned by MAE and JEPA models from the timing/order with which these features are learned, the later being the main focus of this work. Because there is a leap in going between the linear and non-linear setting, we focused on deriving exact results and theoretical guarantees in this paper. Given the long and prolific literature on deep linear models, we view this contribution as self contained, and leave for future work the construction of metrics (of which there may be many) that allow us to analyze the non-linear case where both the features and order are different between the two models.
>
> **Q.** what are the practical implications of the implicit bias observed in JEPA and MAE?
>
> **A.** The main qualitative insight from our results is that noisy (or high variance) features are learned more slowly and with lower amplitude when using the JEPA loss, since a large feature variance across the dataset (denoted as \sigma_i in the paper) reduces the regression coefficient for a fixed cross covariance \lambda_i. A direct prediction that follows from this is that JEPA will tend to focus on the lower subspace of the data variance (PCA space) where most of the perceptual features reside in natural images, as claimed in [1] (see lines 59 - 62 and 74 - 76 in the intro of the paper).
>
> We have conducted additional experiments on ImageNet providing evidence for this claim (see attached pdf). As additional evidence for this in the literature we would like to point the reviewer to [1] which shows how unlike JEPA, reconstruction losses tend to focus on the upper part of the PCA space, and [2] which shows that JEPA tends to learn “slow features” (low variance). Our work can be seen as a first principled analysis of these claims in a toy settings. Additionally, our results provide an intuition to why JEPA objectives are perhaps inefficient for learning features suited for fine-grained pixel level tasks, as those features tend to be noisy (features that would correspond to a low regression coefficient in the linear setting). Finally, our results point to a fundamental limit of the efficiency of the MAE objective in learning semantic features since depth does not meaningfully change the feature learning dynamics (see theorem 4.7 and figure 4 in the paper), unlike the JEPA objective. Questions such as how practitioners should account for these insights and limitations in practice we consider out of scope and left to future work.
>
> [1]: Randall et al: “Learning By reconstruction Produces Uninformative Features for Perception“
>
> [2] Sobal et al: “Joint Embedding Predictive Architectures Focus On Slow Features”
>
>
> We thank the reviewer for their time and effort in reviewing our paper. If we have sufficiently addressed the reviewers concerns, we kindly ask them to consider raising their score.

---

> > ### Comment · Reviewer_5fn3 · 2024-08-13
> > **response by Reviewer 5fn3**
> >
> > I have carefully reviewed the feedback from other reviewers, considered the author’s rebuttal, and followed the ensuing discussion. I appreciate the authors' thorough responses, particularly their additional experimental results (on W1) and answering my questions.
> >
> > Assuming that the insights from these discussions will be included in the final paper, I recommend the paper for acceptance as it provides interesting insights and has the potential to contribute to the ML community and I will raise my score from 5 to 7.

---

### Official Review · Reviewer_Nsp4 · 2024-07-11

**Soundness:** 4
**Presentation:** 4
**Contribution:** 3
**Rating:** 7
**Confidence:** 4

**Summary:**

Analyze learning in two self-supervised paradigms, JEPA and MAE, through the lens of learning dynamics in deep linear networks. Report on a qualitative difference in the order in which features are learned, thus demonstrating their different implicit bias.

**Strengths:**

*	Originality:  the analysis of deep linear networks has been a prolific line of research in terms of deep learning theory, but the application to the self-supervised setting is novel and promising. Related work is adequately cited, as far as I can see.
 *	Quality: the submission is very solid technically, with the assumptions of the theory clearly written and interesting results are derived, demonstrating a clear distinction between the two paradigms.
 *	Clarity: the manuscript is clearly written and well organized, introducing all the relevant background in a concise manner.
 *	Significance: theory provides here a unique view, where the implications of design choices made in development of different algorithms are made visible and thus can educate development of better algorithms. This premise is not fully fulfilled here, see weaknesses.

**Weaknesses:**

*	Quality: this is a theory paper with very limited experimental support (beyond the numerical evaluation of the theory). The random masking and temporal model presented in section 5.1 are very briefly presented, and it is unclear how the simulation results relate to the theoretical predictions. I would expect a clear distillation of an experiment designed to highlight the difference between JEPA and MAE, the qualitative theoretical prediction and how the experimental results support it (or not).
 *	Significance: the implications of the results beyond the theory community are unclear. I would expect the authors to be able to offer (i) some characterization of JEPA and MAE, which is useful for practitioners, or (ii) some classification of possible behaviours of self-supervised algorithms, which can inspire the development of new algorithms with different properties than JEPA and MAE.

**Questions:**

1.	Can you offer qualitative prediction from the theory to real-world systems implementing JEMA or MAE? What would be such prediction if the same deep architecture was trained on both JEMA and MAE loss, what would you expect to see differently in terms of the learned features?
 2.	What is the spectrum of self-supervised algorithms through the lens of deep linear network dynamics?
 3.	What is the result presented in Figure 3? How does it demonstrate a correspondence between theory and the experimental results? Can you offer a similar plot for the random masking task?
 4.	Can you demonstrate the effect of the number of layers is different under JEPA and MAE, as the theory predicts?

**Limitations:**

The authors adequately addressed the limitations of their approach, namely the assumptions about a common diagonalization of the covariance matrices' diagonal dynamics in the low-variance initialization. It would be great if the authors provided experimental support for the applicability of their theory to real-world systems implementing JEPA and MAE or even just provided qualitative predictions derived from the theory.

---

> ### Author Rebuttal · Authors · 2024-08-07
>
> **Q.** “Can you offer qualitative prediction from the theory to real-world systems implementing JEPA or MAE? What would be such prediction if the same deep architecture was trained on both JEPA and MAE loss, what would you expect to see differently in terms of the learned features?”
>
> **A.** (Reproduced from our General Response to All Reviewers): The main qualitative insight from the toy model is that noisy (or high variance) features are learned more slowly and with lower amplitude when using the JEPA loss. A direct prediction that follows from our theory is that JEPA will tend to focus on the lower subspace of the data variance (PCA space) where most of the perceptual features reside in natural images (see lines 59 - 62 and 74 - 76 in the intro of the paper). We have conducted additional experiments on realistic models/data providing evidence for this claim (see attached pdf). As additional evidence for this in the literature we would like to point the reviewer to [1] which shows how reconstruction losses tend to focus on the upper part of the PCA space, and [2] which shows that JEPA tends to learn “slow features” (low variance). Our work can be seen as a first principled analysis of these claims in toy settings. Notably, the toy linear setting not only allows for tractable training dynamics but also has the added benefit that both setups learn the same features. This allows us to focus entirely on comparing the schedule according to which the features are learned for the two setups.
>
> **Q.** What is the spectrum of self-supervised algorithms through the lens of deep linear network dynamics?
>
> **A.** We are not sure we understood the question; could you please clarify what you meant by “spectrum of SSL algorithms”? (Do you mean the linear-algebraic notion of “spectrum”, or are you asking about how the variety of different SSL algorithms manifest in the deep linear setting, beyond MAE and JEPA?)
>
> **Q.** What is the result presented in Figure 3? How does it demonstrate a correspondence between theory and the experimental results? Can you offer a similar plot for the random masking task?
>
> **A.** In Figure 3, we demonstrate that the data distribution considered in our paper (along with its assumed constraints) can in principle be realized — that is, our assumptions are not vacuous. Moreover, it provides a simple data-generative model to help guide intuition about how JEPA and MAE will learn differently in deep linear networks.  We had included two candidates generative processes: one for the static data and the other for time-varying data. Fig 3. focuses on the time-varying (“video-like”) data generation variant. The generative process is given by eq. (17). To summarize, $v^{a}$s are static images, while $u^a=u^a(t)$ are stochastic processes characterized by the autocorrelation coefficients $\gamma^a$s. The Figure 3.a depicts a sample set of $v$s. Fig 3.b shows two sample realizations of the stochastic functions $u^a$s for two different autocorrelation values. Fig 3.{c, d} show the resulting diagonal and off-diagnal pieces of the covariance matrix generated by the process eq.(17). Fig 3e demonstrates how rapidly one converges to the expected values of $\rho$ and $\lambda$ eq (18) derived in the Appendix (as a function of the log of the “mixing time”). The last plot on the right shows diagonal-ity condition is satisfied. These plots together illustrate that the process in eq.(17) generates a data distribution that satisfies our assumptions 4.1.1 and show us how the key parameters of correlation coefficient ($\rho$) and correlation vary with noise and autocorrelation coefficient $\gamma$ in each mode. We do include theory, but not a similar plot for the random masking task (Appendix C.1) . We put the discussion of this in the appendix because we wanted to keep the formulation simple in the main paper and avoid confusing readers, but it is possible to construct a similar formulation for the random masking case with some additional assumptions.
>
> **Q.** Can you demonstrate the effect of the number of layers is different under JEPA and MAE, as the theory predicts?
>
> **A.** Outside of figure 4 which verifies the effect in the linear setting, we have not done comprehensive experiments studying the effect of depth for non-linear networks.
>
> We thank the reviewer for their time and effort in reviewing our paper. If we have sufficiently addressed the reviewers concerns, we kindly ask them to consider raising their score.
>
> [1]: Balestriero et al: “Learning By reconstruction Produces Uninformative Features for Perception“
>
> [2] Sobal et al: “Joint Embedding Predictive Architectures Focus On Slow Features”

---

> > ### Comment · Reviewer_Nsp4 · 2024-08-12
> > **Response to authors**
> >
> > It seems that I had some technical difficulty submitting a response to your rebuttal, and my detailed answer is now lost.
> >
> > In that beautifully written response, I said I was satisfied with your answer, especially with the empirical demonstration of the qualitative prediction in non-linear self-supervised learning, and that I would raise my score (which I did).
> >
> > Also, I tried to explain better what I meant in my puzzling comment on the "spectrum of self-supervised algorithms", but this explanation was not very important just a few hours before the discussion deadline.

---

### Official Review · Reviewer_U4uU · 2024-07-11

**Soundness:** 3
**Presentation:** 3
**Contribution:** 3
**Rating:** 6
**Confidence:** 2

**Summary:**

This paper aims to understand the implicit bias of on two paradigms of self-supervised learning, Joint Embedding Predictive Architectures (JEPAs) and Masked Auto Encoder (MAE). The authors introduce a tractable setting of deep diagonal linear networks and charaterize the learning dynamics of two objectives on the toy problem. Through theoretical analysis, the authors show different behavoirs of two objectives: JEPA prioritizes "influences features" while MAE prioritizes highly-covarying features. These observations are supported numerical experiments on the toy model and the Linear Generative Models.

**Strengths:**

1. This paper is well written the presentation is nice. The symbols and math formulations are clear. The demonstration on Section 3 is helpful on understanding the paper.
2.  The analyis is comprehensive, including both the learning dynamics and the critical time.
3. The difference between the behaviors of JEPA and MAE observed in this paper is interesting.
4. The theoretical results are supported by the numerical experiments.

**Weaknesses:**

Presentation on the setting (Section 2) could be improved. More explanations on $\rho_i, \lambda_i, \sigma_i$ would be helpful to understand the toy setting.

On Section 3, more explanation on the settings are needed, e.g, what's the motivation of choosing these two distributions? What senarios do the distributions represent?

On Section 4, line 165, the subscript $i$ is dropped. This should be said in the beginning of the section to avoid misunderstanding.

The conclusion needs to be clearly elaborated. The authors say that JEPA prioritizes "influential features", whereas MAE prioritizes "highly-covarying features". The connection between "influential/covarying features" and the parameter $\rho_i, \lambda_i, \sigma_i$ is not clear to me.

The study is limited to diagonal linear networks. The analyis on the toy setting provides interesting insights about the learning dynamics of JEPA and MEA. But it's not sure whether the conclusions are applicable to more complex scenarios. I believe it'd be helpful to provide some examples on non-linear cases.

Figure 3 is too small and not anotated.

**Questions:**

Please refer to the "Weakness".

**Limitations:**

Limitations are discussed in the paper.

---

> ### Author Rebuttal · Authors · 2024-08-07
>
> **Q.** “More explanations would be helpful to understand the toy setting.”
>
> **A.** Thank you for pointing this out, we plan to add more elaboration in the rebuttal to make it more accessible.
>
> **Q.** “what's the motivation of choosing these two distributions?”
>
> **A.** The motivation for choosing these distributions is to illustrate how changing the structure of a distribution affects the learning speed of different features, as predicted by our theory. Note that, for a linear model, the only relevant aspects of the training data population statistics are its first and second moments. Hence, when it comes to characterizing data features, \lambda_i and \rho_i are the only relevant quantities to consider (assuming a centered distribution with independent components). In other words, the time it takes to learn the i'th feature has to depend on the feature parameters \lambda_i,\rho_i. The distributions in section 3 are meant to illustrate how varying these feature parameters affects the order of learning for each objective. We will add a note to help clarify this in section 2.
>
> **Q.** “On Section 4, line 165, the subscript is dropped.”
>
> **A.** Thank you for noticing this.
>
> **Q.** “The connection between "influential/covarying features" and the parameter is not clear to me.”
>
> **A.** By “influential features” we simply mean features with a high regression coefficient \rho. In the linear regression literature these features are sometimes referred to as influential/predictive/significant features. By highly covarying features we simply mean features with a high covariance parameter \lambda. Is this explanation clear to the reviewer? We will make this clearer in the intro section, and will include references to this usage in prior work.
>
> **Q.** “...I believe it'd be helpful to provide some examples on non-linear cases.”
>
> **A.** We generally agree with this statement, hence we have conducted the following experiment: one prediction that directly follows from our theory is that the JEPA objective allows the model to focus on the bottom subspace of the observed data variance where, at least for natural images, most of the perceptual features tend to reside [1] (see lines 59 - 62 and 74 - 76 in the intro of the paper). We therefore design an experiment that directly tests this hypothesis (see figures in the added pdf). Additionally, we would like to highlight the fact that beyond analytical tractability, linear models are especially appealing as a testbed in our case for the reason that both the MAE and JEPA objectives eventually learn the same features (perhaps with different amplitudes, see Corollary 4.3 in the paper) if we train the models for sufficiently long. This allows us make fair comparisons on the timescale of learning these features in each objective.
>
> **Q.** “Figure 3 is too small and not annotated”
>
> **A.** Thanks for pointing this out, we will fix it for the final version.
>
> We thank the reviewer for their time and effort in reviewing our paper. If we have sufficiently addressed the reviewers concerns, we kindly ask them to consider raising their score.
>
> [1]: Randall et al: “Learning By reconstruction Produces Uninformative Features for Perception“

---

> > ### Comment · Reviewer_U4uU · 2024-08-09
> > **Acknowledgement**
> >
> > Thank you for the response. My concerns are addressed and I believe this paper would be valuable for the SSL community. Thus I'd raise my score to 6.

---

### Official Review · Reviewer_Mqg6 · 2024-07-13

**Soundness:** 4
**Presentation:** 3
**Contribution:** 4
**Rating:** 7
**Confidence:** 3

**Summary:**

The authors study a characteristic of two common approaches towards visual modeling in SSL--particularly MAE and JEPA. Previous works have demonstrated or identified empirically that JEPA architecture are more prone towards lower variance features whereas MAE optimize towards higher variance features. This work focuses on a theoretical understanding of why this occurs through deep linear encoders. The authors also verify their framework empirically, with both linear and nonlinear encoders (I believe they verify with nonlinear encoders, please see questions for more details).

**Strengths:**

* The theoretical contribution is very important, as the question as to whether raw pixel or feature reconstruction works better is widely debated. Thus, the theory can be used to justify specific architectures in different circumstances.
* The theoretical results are insightful, and the experimental results strongly support the theoretical results.
* The authors experimented with relaxed assumptions, and had the same or similar empirical results, providing evidence that the theoretical results, while derived under unrealistic assumptions, may still hold true.

**Weaknesses:**

* There are very strong assumptions that will almost never be realistic
* I am not convinced that the magnitude of encoders is a proper metric for the training dynamics. There were not many details or justification for this. Please see the questions section for more details.

**Questions:**

* There was no mention of the moving average commonly used in JEPA architectures to update the teacher encoder (the encoder with the stop grad operator). It would be nice to know how different approaches to train/get the teacher encoder affect the training dynamics.
* I am a bit confused by the figures involving the encoder magnitudes, particularly about the values of encoder magnitudes and how they were gotten for each feature. For the first layer, or if the encoder had a single layer, this metric makes sense. However, as linear layers are fully connected, feature i in weight layer 1 does not necessarily correspond strongly with feature i in weight layer 3. Thus, if the average magnitude across each layer for feature i is being used that seems like an invalid metric.
* Is the MLP used in line 478 nonlinear?

**Limitations:**

Yes

---

> ### Author Rebuttal · Authors · 2024-08-07
>
> **Q.** “There was no mention of the moving average commonly used in JEPA...”
>
> **A.**  Indeed EMA (exponential moving average) is often used in practice to boost performance in a variety of SSL methods that employ self distillation, however we argue that the stop gradient operator is the crucial design choice preventing encoder collapse, rather than EMA, which mainly contributes to increased training stability. Indeed, these claims were argued and verified in the SimSiam [1] paper, showing that EMA can be removed altogether without sacrificing performance. Having said that, we agree EMA should be mentioned, and will add a short discussion on it in the paper.
>
> **Q.** “I am a bit confused by the figures involving the encoder magnitudes...”
>
> **A.** Let us clarify. We do not compute the average magnitude across layers. Rather, for each feature i (feature i is the i’th component in the input), we measure the magnitude of the projection of the full encoder on e_i (i’th standard basis), which simply corresponds to the norm of the i’th column of the encoder (see line 127 in the paper). Since we analyze deep linear models, the full encoder is a matrix given by the product of all the layers belonging to the encoder (see \bar{W} in eq 3 in the paper). This makes sense as a metric since the amplitude of the projection of e_i on the encoder is the amplitude of the response to a unit input in feature direction i, which corresponds exactly to how sensitive the encoder latent (output) is to feature i. A zero projection would indicate, for example, invariance to the feature, making it a useless feature for any downstream task that uses the encoder latent.
>
> **Q.** “Is the MLP used in line 478 nonlinear?”
>
> **A.** No, it is a deep liinear MLP
>
> We thank the reviewer for their time and effort in reviewing our paper. If we have sufficiently addressed the reviewers concerns, we kindly ask them to consider raising their score.
>
> [1] Chen et al: “Exploring Simple Siamese Representation Learning”

---

> > ### Comment · Reviewer_Mqg6 · 2024-08-12
> >
> > I have read the reviewer's responses and am satisfied with their rebuttal. They answered all the questions I had.
> >
> > I am keeping my score as a 7 as I believe the paper will have a high impact in the field of Self-Supervised Learning (SSL), helping to theoretically explain the learning phenomena observed in MAE vs. JEPA. While the paper is impactful in systematically explaining this phenomenon, it's not revolutionary in that it was not the first paper to observe and explain this trend. Regardless, having a theoretical backbone is important, and can be used to motivate future research directions within SSL.

---

### Author Rebuttal · Authors · 2024-08-07

**General Response to all Reviewers**

We would like to thank all reviewers for their time and dedication in reviewing our paper, and for their support.
In response to several reviewers, we have conducted additional ImageNet experiments demonstrating phenomena consistent with our theory (described in the attached pdf).
We would also like to address a common question raised by the reviewers in the following general response.

**On the implication of the results to practice:**  The main qualitative insight from our results is that noisy (or high variance) features are learned more slowly and with lower amplitude when using the JEPA loss, since a large feature variance across the dataset (denoted as \sigma_i in the paper) reduces the regression coefficient for a fixed cross covariance \lambda_i. A direct prediction that follows from this is that JEPA will tend to focus on the lower subspace of the data variance (PCA space) where most of the perceptual features reside in natural images, as claimed in [1] (see lines 59 - 62 and 74 - 76 in the intro of the paper). We have conducted additional experiments on realistic models/data providing evidence for this claim (see attached pdf). As additional evidence for this in the literature we would like to point the reviewer to [1] which shows how unlike JEPA, reconstruction losses tend to focus on the upper part of the PCA space, and [2] which shows that JEPA tends to learn “slow features” (low variance). Our work can be seen as a first principled analysis of these claims in a toy settings. Additionally, our results provide an intuition to why JEPA objectives are perhaps inefficient for learning features suited for fine-grained pixel level tasks, as those features tend to be noisy (features that would correspond to a low regression coefficient in the linear setting). Finally, our results point to a fundamental limit of the efficiency of the MAE objective in learning semantic features since depth does not meaningfully change the feature learning dynamics (see theorem 4.7 and figure 4 in the paper), unlike the JEPA objective. Questions such as how practitioners should account for these insights and limitations in practice we consider out of scope and left to future work.

[1]: Balestriero and Lecunl: “Learning By reconstruction Produces Uninformative Features for Perception“

[2] Sobal et al: “Joint Embedding Predictive Architectures Focus On Slow Features”

---

### Decision · Program_Chairs · 2024-09-25

**Decision:**

Accept (poster)

**Comment:**

This paper investigates the implicit biases in two self-supervised learning paradigms: JEPAs and MAE. Through theoretical analysis using deep linear networks, the authors demonstrate that JEPA prioritizes predictive features with minimal noise, while MAE emphasizes high-variance features. These insights are supported by numerical experiments, offering valuable understanding of the distinct feature learning behaviors of JEPA and MAE.

The reviewers (Mqg6, 5fn3, U4uU, MHjv, Nsp4) largely agree that the paper’s theoretical contributions are solid and represent a meaningful addition to the field. Although Mqg6 and MHjv expressed concerns about potentially unrealistic assumptions, the empirical verification under relaxed conditions, as noted by Mqg6, supports the paper’s claims. Additionally, concerns about the limited scope of the experimental results (raised by Nsp4 and 5fn3) were addressed satisfactorily during the rebuttal period with additional experiments.

I recommend accepting this paper for NeurIPS.